# Dynamics of extrachromosomal circular DNA in rice

Jundong Zhuang[1,2,3,4], Yaoxin Zhang [2,3,4], Congcong Zhou[2], Danlin Fan[2], Tao Huang [2], Qi Feng [2], Yiqi Lu[2], Yan Zhao[2], Qiang Zhao[2], Bin Han [2]✉ & Tingting Lu [1]✉

The genome's dynamic nature, exemplified by elements like extrachromosomal circular DNA (eccDNA), is crucial for biodiversity and adaptation. Yet, the role of eccDNA in plants, particularly rice, remains underexplored. Here, we identify 25,598 eccDNAs, unveiling the widespread presence of eccDNA across six rice tissues and revealing its formation as a universal and random process. Interestingly, we discover that direct repeats play a pivotal role in eccDNA formation, pointing to a unique origin mechanism. Despite eccDNA's prevalence in coding sequences, its impact on gene expression is minimal, implying its roles beyond gene regulation. We also observe the association between eccDNA's formation and minor chromosomal deletions, providing insights of its possible function in regulating genome stability. Further, we discover eccDNA specifically accumulated in rice leaves, which may be associated with DNA damage caused by environmental stressors like intense light. In summary, our research advances understanding of eccDNA's role in the genomic architecture and offers valuable insights for rice cultivation and breeding.

The genome, serving as the comprehensive repository of genetic information within an organism, is not a static construct. Instead, it is a dynamic system, ceaselessly undergoing change and adaptation in response to a myriad of internal and external stimuli[1,2]. Despite notable advances in genomics, the genome's dynamic nature are in mystery[3]. Unraveling these enigmas is vital for our understanding of life mechanisms and evolutionary biology.

One such enigmatic component of the genome is extrachromosomal circular DNA (eccDNA), a type of DNA that exists outside the primary chromosomes[4,5]. In 1965, it was first reported that there were paired eccDNAs called double minutes in several eukaryotic organisms including wheat germ, boar sperm and human tumor specimens[6,7]. In 1978, researchers identified a mechanism of action for eccDNA by finding that double minutes in methotrexate-resistant mouse cells carried the *DHFR* gene, which mediated the cells'

resistance to the drug[8]. Since then, eccDNA has been found to be common in many types of tissues in humans, both healthy and malignant[9]. Recently, its contribution to the dynamic nature of the genome is increasingly being recognized[4,5,10,11]. The proposed roles of eccDNAs within the genome are manifold, ranging from gene amplification, regulation of gene expression, to contributions to genetic variability[4,12]. However, research on eccDNA predominantly focuses on human biology, and its exploration in other species, especially in the realm of botany, is scant[9,13,14]. Using two-dimensional gel electrophoresis, some eccDNAs have been identified in dicotyledonous plants such as Arabidopsis and daisies, suggesting that eccDNA may also be present in plants more generally[15]. It has been proposed that eccDNAs may play a role in the evolution of satellite repeats[16]. As such, our understanding of eccDNA in plants, its genesis, distribution, and functional implications, remains nascent[17,18].

[1]School of Life Sciences and Biotechnology, Shanghai Jiao Tong University, Shanghai 200240, China. [2]National Center for Gene Research, National Key Laboratory of Plant Molecular Genetics, CAS Center for Excellence in Molecular Plant Sciences, Institute of Plant Physiology and Ecology, Chinese Academy of Sciences, Shanghai 200032, China. [3]University of Chinese Academy of Sciences, Beijing 100039, China. [4]These authors contributed equally: Jundong Zhuang, Yaoxin Zhang. ✉e-mail: bhan@ncgr.ac.cn; xyzltt@sjtu.edu.cn

In this work, we conduct a comprehensive investigation into the presence and role of eccDNA within the rice (*Oryza sativa* L.) genome, a staple food of global importance and China's principal crop. Our research aims to elucidate the contribution of eccDNA to the inherent dynamism of the genome, focusing on its distribution, characteristics, and the functional significance in rice. Special emphasis is placed on understanding the formation process of eccDNA and its potential implications for chromosomal integrity. Additionally, we uncover and detail an interesting observation: rice leaf tissue exhibits notably higher quantities of eccDNA, with light exposure identified as a primary influencing factor. The findings from our study promise to enhance our understanding of genomic integrity, offering avenues to boost crop resilience and yield in response to environmental challenges.

## Results

### Characteristics of eccDNA across rice tissues

In this study, we aimed to identify and analyze eccDNA in various rice tissues. Utilizing the Circle-seq method originally developed by Møller[19], we extended its application to the plant domain. This method entails enzymatic degradation of linear DNA, enriching for eccDNAs resistant to exonuclease activity, thereby enhancing their genome-wide detection (Fig. 1a). Specifically, we conducted Circle-seq analyses on six distinct rice tissues: germinated embryos, roots, stems, leaves, leaf sheaths, and panicles (Supplementary Fig. 1). We prepared two biological replicates for each tissue type, resulting in a total of 12 sequenced samples. These samples collectively yielded approximately 28 billion high-quality, paired end reads. Each of these reads uniquely aligned with the rice reference genome (MSU v7), as detailed in Supplementary Data 1.

Using our analysis method (refer to Supplementary Note 1 for detailed methodology), we reveal a significant presence of eccDNAs across all 12 samples, spanning six distinct tissue types (Fig. 1b; Supplementary Data 2). Despite the high frequency of eccDNAs within each sample, there was scarcely any overlap between them. This observation implies that the eccDNA profiles are highly unique to each sample, even among those from the same type of tissue (Fig. 1c, d; Supplementary Data 3–5). The observed diversity in eccDNA patterns across identical tissue types may be influenced by several factors, such as the random processes governing eccDNA formation, the molecular diversity within tissues, and the precision of our detection methods. Together, these factors likely shape the unique eccDNA landscapes evident in each sample, highlighting the stochastic nature of eccDNA production.

To further explore this distribution, we combined eccDNAs from replicate samples of each tissue. A total of 25,598 eccDNAs were identified across six tissues (Fig. 2a). We observed significant variation in the quantity of eccDNA among different tissues. Interestingly, an astonishing 86.66% of eccDNA was found in the leaves, possibly hinting at an unusual relationship between rice leaves and eccDNA. An astounding total of 22,183 eccDNAs were identified in the leaf samples, comprising 86.66% of the overall eccDNA count. This finding indicates a potentially unique and significant association between eccDNAs and the leaf tissue in rice. Additionally, we found that the GC content of eccDNAs, as well as in the 150 bp regions flanking them, was higher than the average across the entire rice genome, regardless of tissue type (Fig. 2b). Particularly, the GC content in eccDNAs was substantially higher. Although there were differences in the quantities of eccDNAs among tissues, their length distribution was relatively

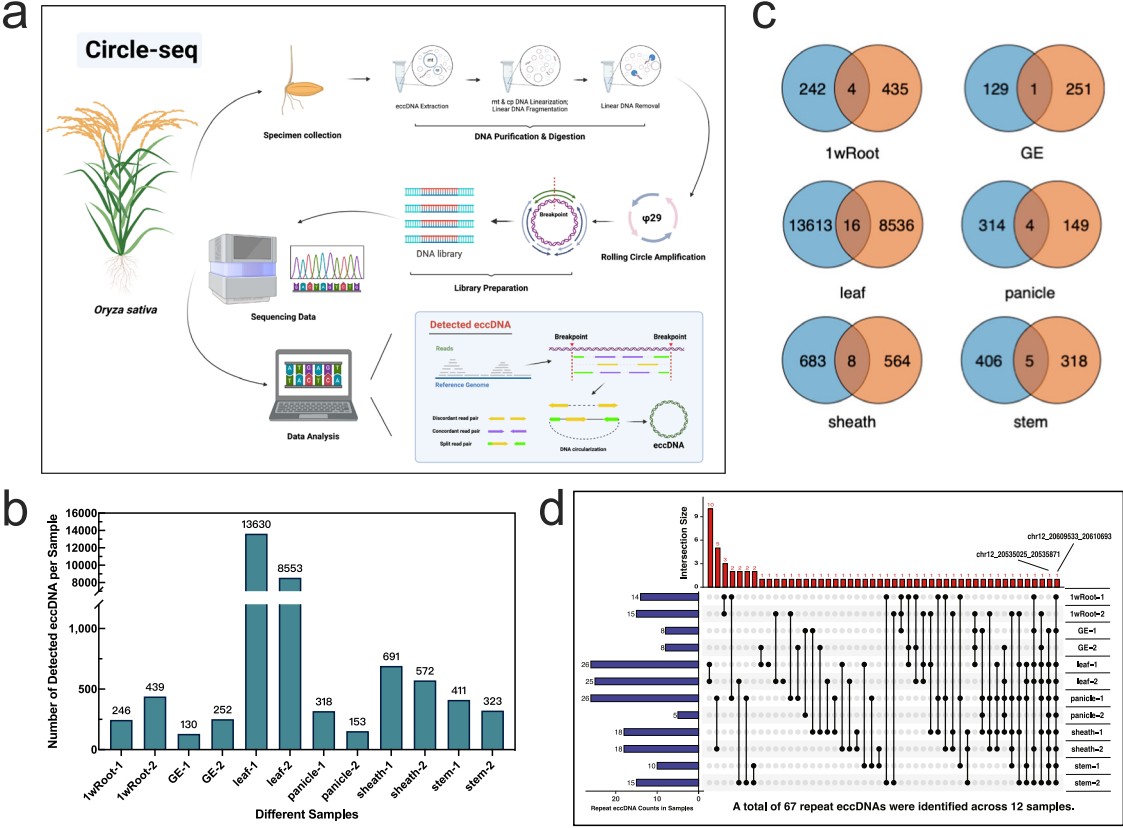

**Fig. 1 | Detection of eccDNAs in six different rice tissues (12 samples) using Circle-seq. a** Schematic representation of the Circle-Map pipeline for eccDNA detection. Created with BioRender.com. **b** The identified eccDNAs in the six tissues. For each tissue, two biological replicates were prepared. **c** A series of Venn diagrams displaying the overlaps of eccDNA populations between the two biological replicates from the same tissue. **d** Detailed exploration of eccDNA repeat occurrences among the samples, underscoring the diversity and intricacy of eccDNA profiles. Source data are provided as a Source Data file.

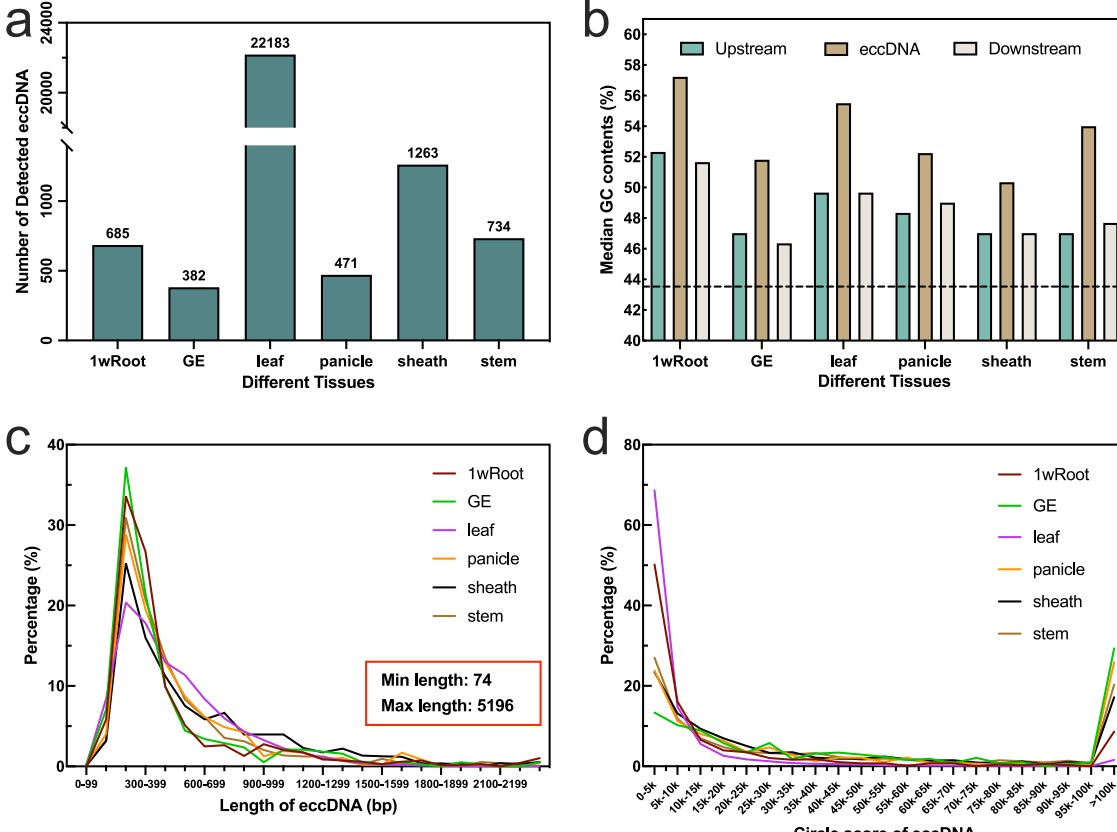

**Fig. 2 | Characteristics of eccDNAs across six rice tissues. a** Distribution of eccDNA counts across different rice tissues. **b** Comparison of GC content in eccDNAs with genomic sequences located 150 base pairs (bp) upstream and downstream of the eccDNA source loci. The average GC content of the rice genome is depicted by the black dotted line. **c** Length distribution of eccDNAs in the various tissues, ranging from 74 bp (minimum) to 5196 bp (maximum). **d** Distribution of Circle Scores for eccDNAs across different rice tissues. Source data are provided as a Source Data file.

consistent (Fig. 2c). From the data, it's apparent that most samples had the highest percentage of eccDNAs in the 200–400 bp range, with a general trend of decreasing percentages as the length increased. This trend was consistent across all samples, even in leaf tissue with its high content of eccDNA, suggesting that shorter eccDNAs are more common in these tissue samples. Approximately 49.74% of eccDNAs were within the 200–400 bp range.

Considering that Circle-seq, due to the non-linear amplification inherent in the use of phi29 polymerase, does not provide a strictly quantitative measure of circle abundance, we used a Circle score calculated with Circle-Map software to assess the quality of eccDNA. The Circle score is an additive scoring scheme that takes into account the alignment quality, length of the split segment, and the number of split reads supporting the circular DNA. As shown in Fig. 2d, the line graph displays the distribution proportions of detected eccDNAs in different plant tissue samples. For most samples, the proportion of detected eccDNAs significantly decreased as the Circle score increased. In the low Circle score range (i.e., from 0 to about 20k), the proportion of detected eccDNAs was higher, while in the middle range (20k–100k), this proportion dropped to a very low level. However, in the highest Circle score range (over 100k), all samples seemed to show a slight increase, indicating the presence of a smaller quantity of high Circle score eccDNAs in these samples.

### Formation mechanisms of eccDNA in rice

To investigate eccDNA formation, we examined sequence features at the termini of eccDNA, expanding both ends outward by 100 bp and inward by 50 bp. Through BLAST comparison, we identified direct repeats (4.40%) and inverted repeats (0.16%) within the eccDNA

(Fig. 3a, b). To assess the randomness of this observation, we developed a model comprising six datasets, each containing 1000 random theoretical eccDNA sequences. Our analysis revealed that the proportion of eccDNA with direct repeat sequences significantly exceeded predictions from our model (Fig. 3c). This observation suggests that, despite the relatively low frequency of direct repeat eccDNAs (hereinafter referred to as DR-eccDNA), their presence is non-random, hinting at a potentially crucial role in specific processes. Moreover, we observed that approximately 95% of the direct repeat fragments within these eccDNAs range from 7–15 bp in length (Fig. 3d).

Further analysis revealed that, compared to eccDNAs lacking direct repeats (hereinafter referred to as Other-eccDNAs), DR-eccDNAs have a significantly lower GC content but exhibit higher mass and greater length (Fig. 3e–h). This pattern suggests that DR-eccDNAs might be more stable than those with higher GC content. Additionally, a comprehensive analysis using Hedges' g effect sizes revealed distinct characteristics between DR-eccDNAs and Other-eccDNAs. Specifically, the DR-eccDNA group showed a notably lower GC content compared to the Other-eccDNA group, with an effect size of -0.560, indicating a moderate-to-large difference in mean values. This significant reduction in GC content might suggest variations in stability or functionality between the groups, where the lower GC content in DR-eccDNAs could be linked to increased stability or distinct biological roles. Conversely, the DR-eccDNA group demonstrated a significantly greater eccDNA length, with an effect size of 1.134, emphasizing a large and statistically significant difference. This finding suggests that DR-eccDNAs tend to be larger, which might reflect differences in their genomic architecture or imply a different origin or formation mechanism compared to Other-eccDNAs. Furthermore, the Circle Score, with a smaller effect

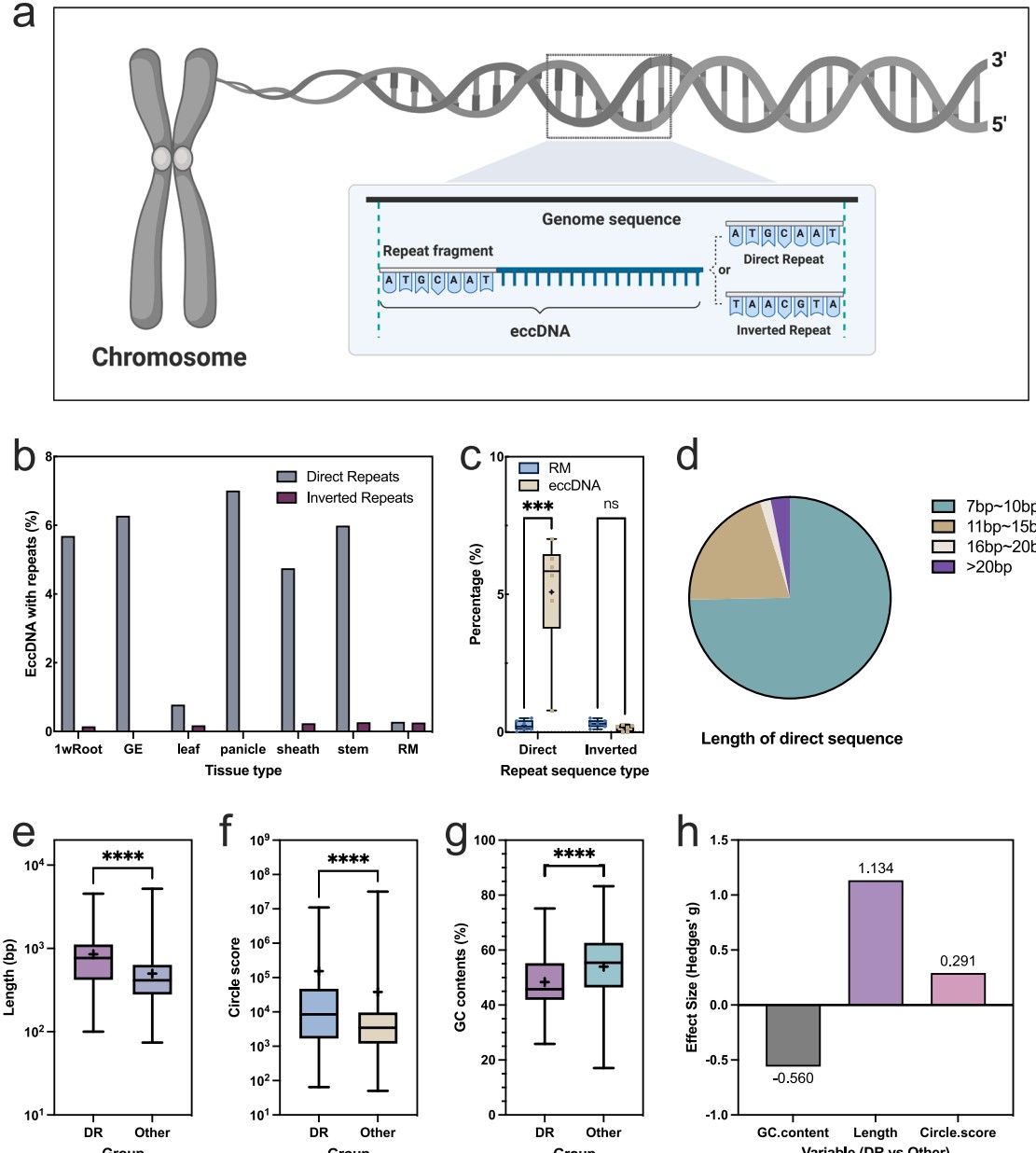

**Fig. 3 | Direct repeats contribute to the formation of high quality eccDNA.**
**a** Depicts the correlation between the presence of direct and inverted repeats in the genesis of eccDNA. Created with BioRender.com. **b** Conducts a proportional analysis contrasting direct versus inverted repeats within eccDNAs extracted from assorted rice tissues against sequences generated at random. **c** Box-and-whisker plots compare direct and inverted repeat prevalence in eccDNAs and a Random Model (RM) across six biologically independent samples, with individual data points shown. Statistical significance for direct repeats is marked ($p = 0.000349$, adjusted $p = 0.000699$), not significant for inverted repeats ($p = 0.058097$), using a two-tailed Unpaired $t$ test with df = 10. **d** Depicts the length distribution of direct repeats within eccDNAs. **e** Contrasts the lengths of DR-eccDNAs ($n = 373$, range: 100–4542 bp, median: 767 bp) against Other-eccDNAs ($n = 25,345$, range: 74–5196 bp, median: 414 bp), showing a notable difference ($p < 0.0001$). **f** Assesses

the Circle Score quality of DR-eccDNAs in comparison to Other-eccDNAs, indicating superior quality in DR-eccDNAs ($p < 0.0001$). **g** Compares the GC content in DR-eccDNAs ($n = 297$, range: 25.87–75.16%, median: 45.77%) with Other-eccDNAs ($n = 25,345$, range: 17.09–83.27%, median: 55.38%), identifying significant variances ($p < 0.0001$). **h** Applies Hedges' g to measure effect size in eccDNA comparisons. The $p$ values for (**e**, **f**, **g**) were calculated using a two-sided Mann–Whitney test. Data representation includes mean values denoted by a "+" symbol to indicate the central tendency, with box plots elucidating the distribution featuring minimum, maximum, median, 25th (Q1), and 75th (Q3) percentiles for detailed analysis. Statistical significance is highlighted by p values, with asterisks denoting significance levels (*$p < 0.05$, **$p < 0.01$, ***$p < 0.001$, ****$p < 0.0001$). Source data are provided as a Source Data file.

size of 0.291, still indicates a positive difference favoring the DR-eccDNA group. Although this effect is modest, it may represent a meaningful distinction in the circularity or structural integrity of the eccDNAs, potentially influencing their cellular persistence and function. Collectively, these findings highlight the unique features of DR-eccDNAs in comparison to Other-eccDNAs, with implications for their stability and function that warrant further investigation.

We further validated our findings using PCR on eight different tissues, with both convergent and divergent primer pairs (Fig. 4a). To circumvent potential artifacts arising from enrichment steps, we opted for unamplified, raw eccDNA samples as PCR templates. Interestingly, we consistently identified eccDNAs (chr12_20609533_20610693 and chr12_20499833_20500504) across all samples, albeit with variable signal intensities (Fig. 4b,c). These results underline the validity of our

experimental approach and establish eccDNA formation as a universal cellular event.

We then sought to unequivocally confirm the circular nature of eccDNAs, given that genomic rearrangements on a single chromosome can produce visible head-to-tail junction sequences which, despite their appearance, remain linear. To this end, we subjected eccDNA samples to treatment with DNA exonuclease, an enzyme that selectively degrades linear DNA while leaving circular DNA intact. Our PCR analyses conducted on samples both pre- and post-exonuclease treatment revealed complete digestion of the linear control (*Actin1*), with the eccDNA remaining predominantly undigested (Fig. 4d). This outcome unequivocally demonstrates the circular configuration of eccDNAs, a conclusion we further reinforced by performing the exonuclease assay on stem samples as well (Fig. 4e).

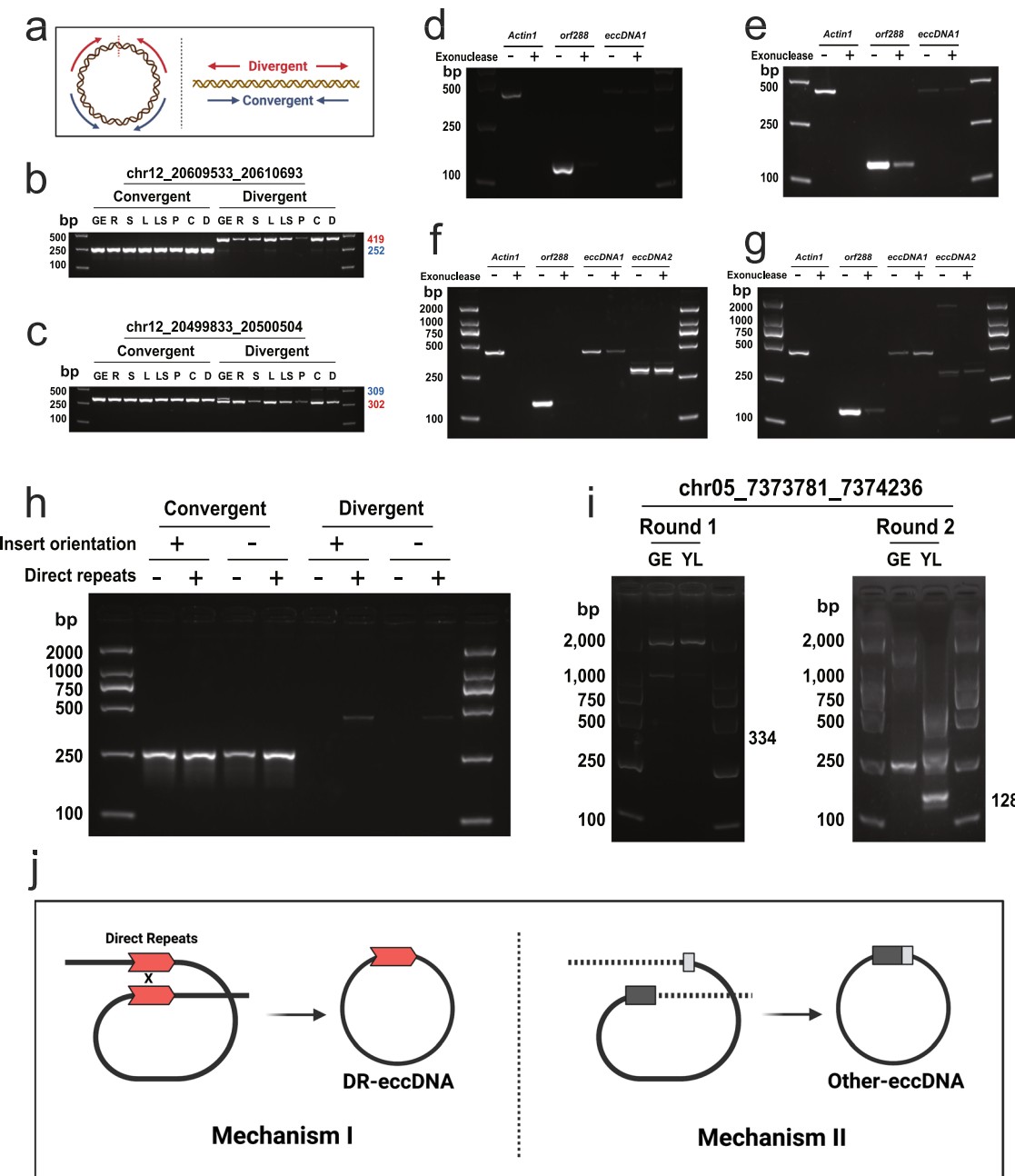

**Fig. 4 | Two kinds of potential mechanisms for the formation of eccDNA. a** The PCR validation method used, featuring both convergent and divergent primer pairs. **b, c** The consistent identification of eccDNAs chr12_20609533_20610693 (eccDNA1) and chr12_20499833_20500504 (eccDNA2) across tissue types: germinated embryos (GE), roots (R), stems (S), leaves (L), leaf sheaths (LS), and panicles (P). **d** Exonuclease treatment analysis on germinated embryo samples showing exonuclease resistance of eccDNA1, with linear DNA control (Actin1) and orf288 demonstrating susceptibility, indicating the circular structure of eccDNA1. **e** Confirmation of eccDNA1's circularity through comparable exonuclease resistance in stem samples. **f** Exonuclease assessment of leaf sheath samples for Actin1, orf288, eccDNA1, and eccDNA2. **g** Parallel exonuclease analysis in panicle samples reinforces the circularity evidence for eccDNA1 and eccDNA2. The expected amplicon sizes of Actin1, orf288, EccDNA1 and EccDNA2 are 423, 135, 419 and 302 bp, respectively. **h** The role of direct repeats in eccDNA genesis examined via construct experiments. **i** Detection of an Other-eccDNA (chr05_7373781_7374236) through nested PCR assays. **j** A bifurcated model of eccDNA formation in rice, highlighting the differences in mechanisms and properties between DR-eccDNAs and Other-eccDNAs. Created with BioRender.com. Given the stochastic nature of eccDNA formation, each experiment was independently conducted only once. Source data are provided as a Source data file.

Additionally, we scrutinized another eccDNA, chr12_20499833_20500504, in leaf sheath samples (Fig. 4f) and panicle samples (Fig. 4g). Sanger sequencing of the PCR products of chr12_20499833_20500504 showed two distinct amplicons, corresponding to two variants of eccDNA with a 7-bp difference within the 34-bp direct repeat sequence (Supplementary Fig. 2). Furthermore, validation was extended to another set of ten eccDNAs (Supplementary Fig. 3), each containing a short sequence homologous to sequences neighboring their genomic loci.

A strong correlation was observed between the length of the direct repeats and the signal strength of divergent PCR products. Our construct experiments further highlighted the role of direct repeats in eccDNA formation, regardless of minor variations (Fig. 4h). In contrast, Other-eccDNAs, such as chr05_7373781_7374236, were only detected through a nested PCR assay (Fig. 4i). Intriguingly, Sanger sequencing of the PCR product identified two distinct eccDNAs, neither of which matched the originally identified sequence (Supplementary Fig. 2). This led us to postulate that eccDNAs lacking direct repeats may not consistently arise from the same genomic segment.

Our deeper understanding of these processes has led to the proposition of a bifurcated model of eccDNA genesis in rice (Fig. 4j). In this model, DR-eccDNAs are primarily generated via a certain biological process. Although these DR-eccDNAs are less abundant, their inherent stability makes them more detectable in rice, indicating a robust yet less frequently utilized pathway of formation. In contrast, Other-eccDNA are predominantly brought into existence through a distinct process. Despite their greater number, these eccDNAs exhibit considerable randomness, suggesting a more stochastic and prolific mechanism of formation. This dichotomy not only reveals the diverse strategies employed by the organism for eccDNA formation, but also underscores the intricate complexity of the genomic landscape.

## The impact of eccDNA on rice genome stability

Building upon this understanding of eccDNA genesis, we proceed to explore the consequential impacts on the chromosomal landscape. As eccDNA formation inherently involves the excision of chromosomal material, it could introduce chromosomal deletions. To thoroughly examine this, we leveraged the capabilities of both Next-Generation Sequencing (NGS) and Third-Generation Sequencing (TGS), employing these techniques on three distinct rice samples—two samples subjected to NGS, and one sample to TGS. In an exploration of the resequencing data, we discerned two distinct classes of sequencing reads in specific regions of the reference genome (Fig. 5a). The first class of reads exhibited an impeccable alignment with the reference genome. The second class, characterized by a partial absence of the eccDNA sequence, achieved only an incomplete alignment with the reference genome.

The results indicate that chromosomal microdeletions caused by eccDNA are more readily detected by TGS technologies (Fig. 5b). Specifically, in the region associated with eccDNA (chr03_28137649_28137850), we detected five reads in the TGS dataset that aligned flawlessly with the reference genome. Moreover, we observed 12 reads in the TGS data that conspicuously lacked the eccDNA sequence, underscoring their absence and further emphasizing the precision of TGS technology in detecting such deletions (Fig. 5c).

We then aimed to detect a microdeletion associated with the formation of chr12_20609533_20610693, an eccDNA across all 12 samples. PCR assays confirmed the presence of both the microdeletion and the intact sequence (Fig. 5d). Sanger sequencing validated that the PCR products corresponded to chromosomal fragments devoid of the entire eccDNA sequence. The local retention of a single direct repeat, left behind by the formation of DR-eccDNA, further substantiates the premise that eccDNA genesis can induce minor chromosomal deletions (Supplementary Fig. 4). These findings not only confirm that minor chromosomal deletions can be a result of eccDNA formation, but also imply that the process of eccDNA generation may function, at least in part, independently from traditional replication mechanisms (Fig. 5e).

## Distribution of eccDNA across the chromosomal landscape in rice

In our study, we investigated the distribution patterns of eccDNA across different chromosomes. As illustrated in Fig. 6a, b, our findings suggest that eccDNAs are distributed both uniformly and randomly along the chromosomes. There appears to be a strong direct correlation between the quantity of eccDNA and chromosomal length, as indicated by a correlation coefficient ($R$) of 0.97, with a $p$ value of 1.33e−07.

Further in-depth analysis reveals a significant enrichment of eccDNA in intergenic regions, while correspondingly, there is a notable reduction in the proportion of eccDNA within genetic areas (Fig. 6c). What's interesting is that our examination of the relationship between eccDNA and different genomic regions showed a notable enrichment of eccDNA within coding sequences (CDS). In contrast, introns and 3′ untranslated regions (3′ UTR) exhibited a significantly lower presence of eccDNA. Interestingly, microRNA (miRNA) regions demonstrated substantial eccDNA enrichment. We also explored eccDNA's association with repetitive elements in the genome. Our results indicate a marked decrease in eccDNA within LINE and RC regions, while LTR, DNA, and Retroelement repeat sequences showed a notable increase in eccDNA presence. Moreover, we categorized eccDNAs based on their Circle score into two groups and investigated their potential preference for genomic regions. The findings indicate that these two categories of eccDNA exhibit virtually no genomic preference (Supplementary Fig. 5).

In exploring the interactions between eccDNA and genes, we found that in samples other than leaf tissues, approximately 40% of eccDNAs overlapped with at least one gene (Fig. 6d). Additionally, over 99% of the eccDNA instances were associated with a single gene, while less than 1% spanned across two genes (Fig. 6e). Most genes overlapping with eccDNA were linked to a single eccDNA molecule (Fig. 6f). Intriguingly, a small number of genes overlapped with multiple eccDNAs; for instance, the gene *LOC_Os12g33958* overlapped with 36 different eccDNAs, and *LOC_Os08g22790* overlapped with 14 different eccDNAs. In further analyses, we categorized eccDNAs containing genes as 'GcE' and genes containing eccDNAs as 'EcG'. This revealed that most genes were classified as GcE, with only 85 eccDNA instances encompassing entire genes (Fig. 6g). This might be related to the length of eccDNAs, considering that about 50% of eccDNAs are between 200 and 400 bp in length. Lastly, we conducted Gene Ontology (GO) enrichment analysis on the genes overlapping with eccDNAs in six different tissues, focusing on the enriched biological processes (BP). We identified the unique BP GO terms enriched in each tissue and found commonalities among them (Fig. 6h). By analyzing these 18 BP GO terms shared across the six tissues, we discovered that these genes predominantly participate in biological stress responses and cellular metabolism (Fig. 6i).

## The relationship between eccDNA and gene in rice

To analyze whether eccDNA influences gene expression, we performed RNA sequencing (RNA-seq) on all 12 samples, examining gene expression patterns across six different tissue types in rice plants. We extracted expression values (TPM) of genes overlapping with identified eccDNA from these samples and obtained expression values for the same gene in another replicate sample from the same tissue type. A paired $t$ test revealed no significant difference in gene expression between samples with and without gene-overlapping eccDNA (Fig. 7a). This suggests a lack of direct correlation between eccDNA and gene expression.

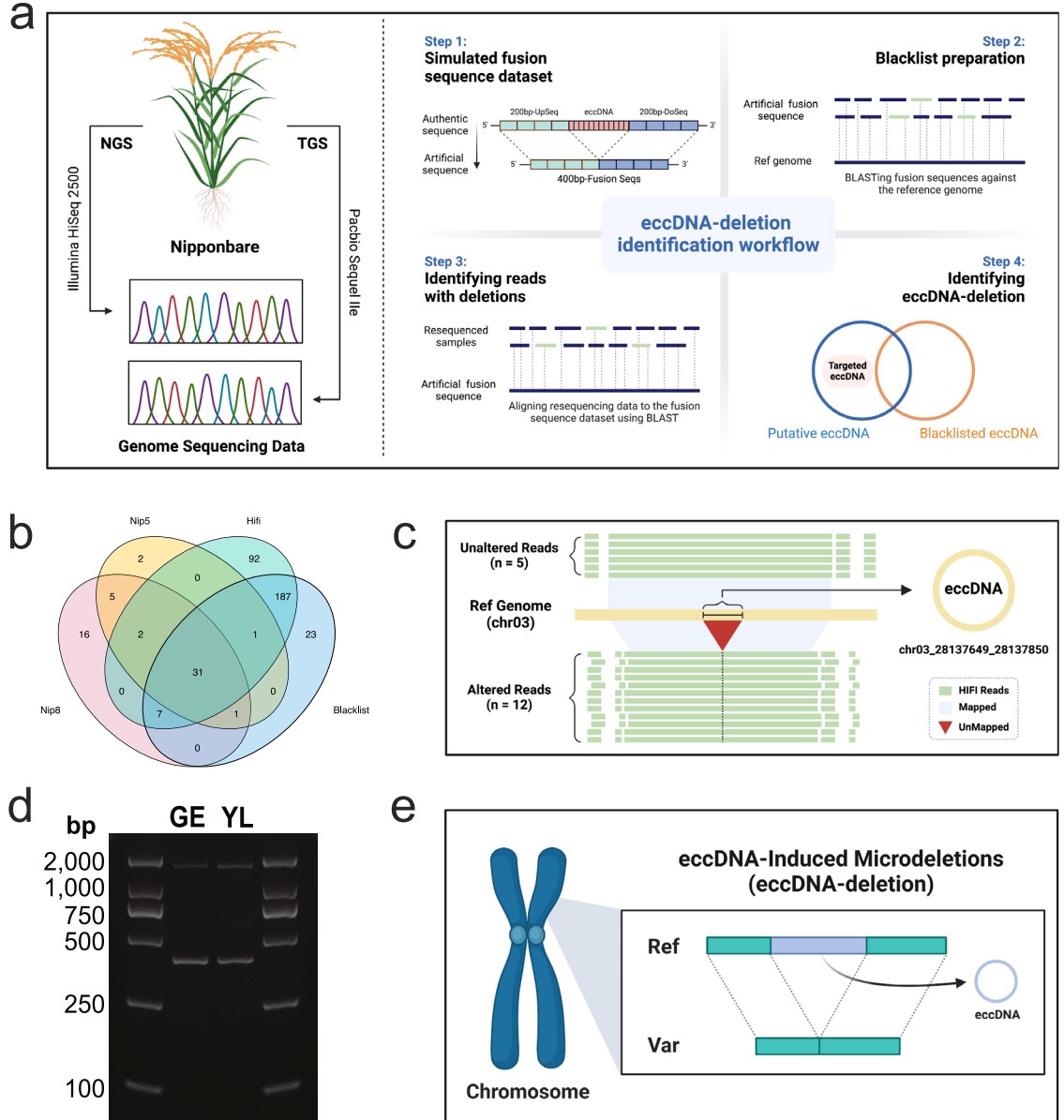

**Fig. 5 | The influence of eccDNA on the stability of the rice genome. a** Illustration of the experimental approach using Next-Generation Sequencing (NGS) and Third-Generation Sequencing (TGS) on three distinct rice samples. Created with BioRender.com. **b** The Venn diagram of the results of three samples. **c** An instance within the eccDNA-associated region (chr03_28137648_28137851), showing five reads perfectly aligned with the reference genome and another 12 reads missing the eccDNA sequence. Created with BioRender.com. **d** An agarose gel image of PCR detection of a chromosomal microdeletion corresponding to chr12_20609533_20610693, with the expected PCR product sizes of deleted and non-deleted chromosomes indicated on the right. The tissues used for DNA preparation are indicated as follows: germinated embryo (GE) and young leaf (YL). Given the stochastic nature of eccDNA formation, the experiment was conducted only once. **e** A schematic diagram showing that minor chromosomal deletions can result from eccDNA formation. Created with BioRender.com. Source data are provided as a Source Data file.

Previous analysis indicated a propensity for eccDNA to enrich in regions associated with stress-responsive genes (Fig. 6h, i). To explore the relationship between eccDNA and stress conditions, we prepared two sets of samples, each comprising two normal Nip rice seedlings and two subjected to drought stress. We then analyzed their eccDNA profiles. Compared to the normal group (Nip), the drought stress group (Dro) showed a slight increase in eccDNA counts, but the difference was not statistically significant (Fig. 7b; Supplementary Data 6). There were also no significant differences in GC content, length distribution, or circle score of eccDNA between the two groups (Fig. 7c–e). The genomic distribution of eccDNA was similar in both sample sets. Interestingly, an analysis of eccDNA's genomic distribution revealed a significant association with the 5′ UTR regions of genes (Fig. 7f). Furthermore, the proportion of stress-responsive genes overlapping with eccDNA remained almost constant across different samples (Fig. 7g). Overall, these findings suggest that stress conditions, particularly drought stress, do not directly influence the production of eccDNA. The underlying reasons for eccDNA's tendency to enrich in regions associated with stress-responsive genes warrant further analysis and exploration.

**High abundance of eccDNA in rice leaf tissues**
In our study, eccDNA in rice leaf samples exhibited a pattern distinct from other tissues. Specifically, leaf samples contained a larger

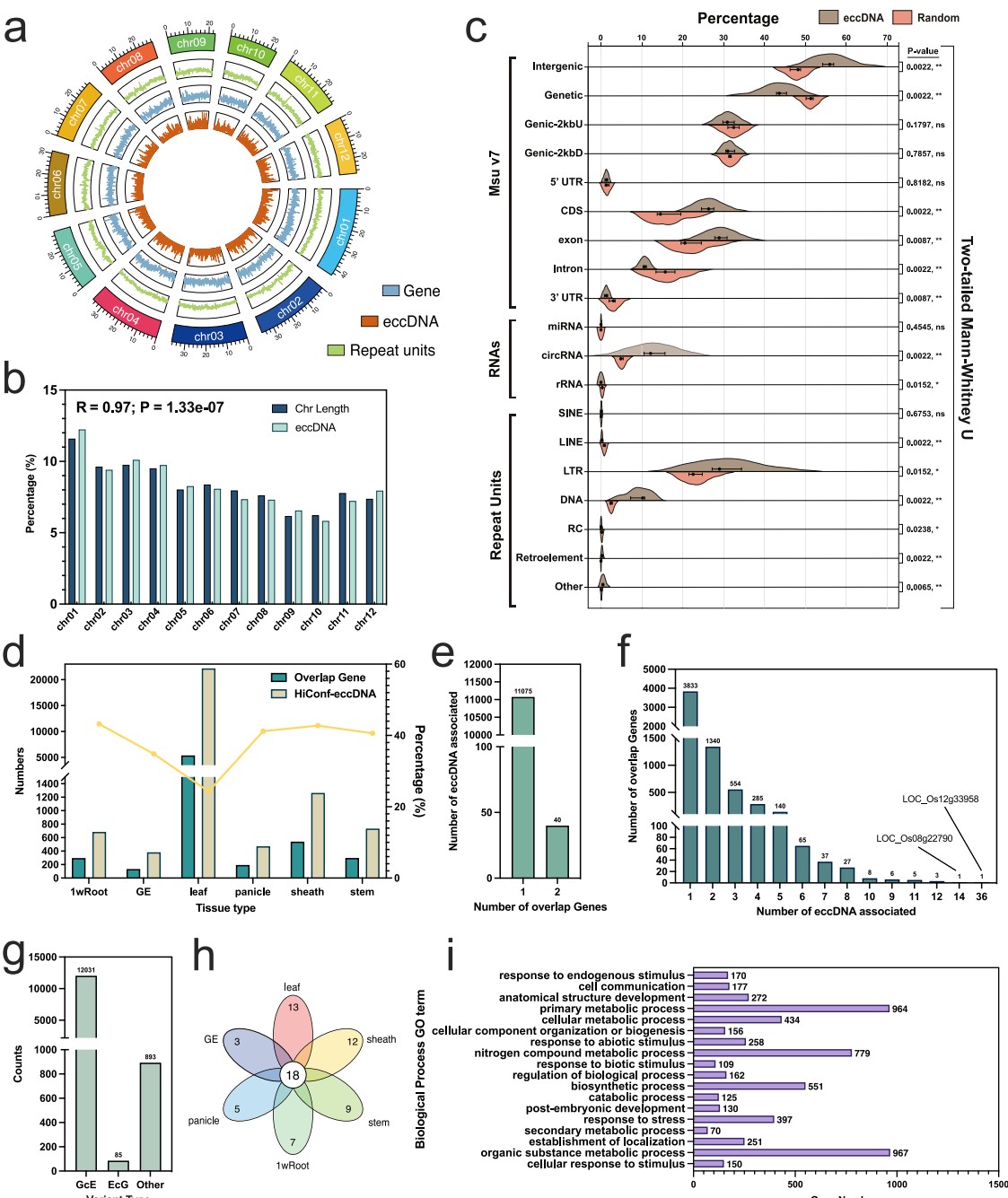

**Fig. 6 | The distribution of eccDNA across chromosomal territories and its relationship with various gene regions. a** The uniform and random distribution of eccDNA across chromosomes. **b** A direct correlation between the quantity of eccDNA and the chromosomal length. **c** The association between eccDNA and various genomic regions with six data points per group for both eccDNA and Random. Statistical significance is highlighted by *p* values, with asterisks denoting significance levels (*$p < 0.05$, **$p < 0.01$, ***$p < 0.001$, ****$p < 0.0001$). **d** The proportion of genes overlapping with eccDNA. **e** The percentage of eccDNA associated with a singular gene, and eccDNA spanning two genes. **f** The predominant type of overlapping genes associated with a single eccDNA molecule. **g** Comparison of gene-containing eccDNA (GcE) and eccDNA-containing genes (EcG). **h** Venn diagram of BP GO terms enriched in genes overlapped by eccDNA across six unique tissues. **i** A list of the shared BP GO terms among these genes, primarily involved in biological stress and cellular metabolism. Source data are provided as a Source Data file.

number of eccDNAs, most of which had a lower proportion overlapped with genes. We hypothesize that this phenomenon might be unique to rice leaves. To test this hypothesis, we analyzed eccDNA samples from *Arabidopsis thaliana* (across four tissues, with three samples each), drawing on data from Wang, et al.[17]. The results indicated that while the eccDNA content varied significantly among different tissues of *Arabidopsis thaliana*, there was no marked increase in eccDNA content in its leaves (Fig. 8a). This observation contrasts with studies in mammals,

where tumor regions often contain significantly more eccDNA than other tissues—a hallmark of the tumor area. Thus, we infer that the phenomenon of high quantity eccDNA might be specific to rice leaf tissues. To further investigate the origin of these eccDNAs in rice leaves, our analysis revealed that only about 1.7% of leaf eccDNAs could be mapped to organellar genomes, predominantly the mitochondrial genome (Fig. 8b; Supplementary Data 7). Moreover, we specifically identified and distinguished mitochondrial eccDNAs from those

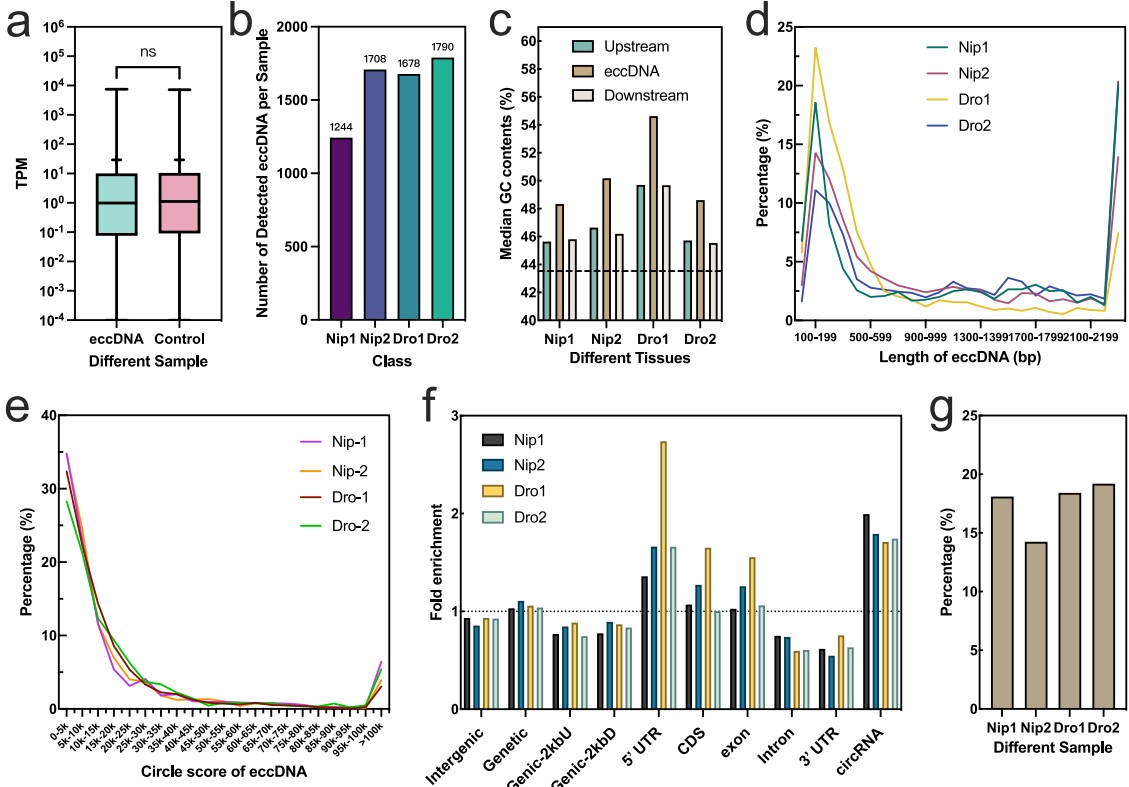

**Fig. 7 | Interaction of eccDNA with gene expression patterns and stress conditions. a** Comparison of Gene Expression in Samples with and Without eccDNA Overlap. This figure presents the proportion of genes with low expression levels in two distinct sample types: those with gene-eccDNA overlap (Goe) and control samples without such overlap. Analyzing 3593 gene pairs using a two-tailed paired $t$ test indicated no significant difference in expression levels between the groups ($p = 0.8192$, df = 3592). Data representation includes mean values denoted"b" a "+" symbol to indicate the central tendency, with box plots elucidating the distribution featuring minimum, maximum, median, 25th (Q1), and 75th (Q3) percent"le". "ns" signifies no significant difference. **b** The count of eccDNA between two groups of Nip rice seedlings, one under normal conditions and the other under drought stress (Dro). **c** The GC contents of eccDNA between the two aforementioned groups. **d** The length distribution of eccDNA between the two grIs. **e** The circle score distribution of eccDNA between the two groups. **f** The enrichment of four sample eccDNAs in the indicated genomic region relative to the expected percentage based on random distribution. **g** The proportion of stress genes overlapped by eccDNA across different samples. Source data are provided as a Source Data file.

originating in chloroplasts (Fig. 8c). The results showed that the quantity of mitochondrial eccDNAs did not significantly differ across various rice tissues. Interestingly, even for chloroplast-related eccDNA, the leaves contained only 12 instances. Although this number was higher than in other tissues, the difference was not statistically significant when compared to other tissue types.

Considering that rice is a plant requiring intense sunlight and acknowledging that ultraviolet (UV) light can cause chromosomal breakage, we hypothesized that the abundance of eccDNA in rice leaves might be linked to the light required for their growth. To explore this, we conducted additional experiments using 10-day-old samples of the Nip variety under three different conditions: darkness (28 °C), control, and UV irradiation (29 °C, with 13 h of light/11 h of darkness). For the UV treatment, the samples were exposed to 78μW/cm² for 20 min. Our results indicated a significant increase in eccDNA abundance in leaf tissue post-UV treatment compared to the control group (Fig. 8d; Supplementary Data 6). Intriguingly, eccDNA levels in the darkness group were also higher than those in the control. Given that darkness represents a stress condition for rice, the increase in eccDNA under these conditions seems logical. This stress-induced increase suggests that both darkness and UV light can influence eccDNA accumulation, with the significant rise in eccDNA levels following UV exposure being the most notable observation. It underscores UV light as a primary inducer of eccDNA formation. These results provide preliminary evidence supporting our hypothesis that light conditions, particularly UV exposure, significantly influence

eccDNA formation in rice leaf tissue. However, to elucidate the deeper mechanisms underlying this phenomenon, further detailed investigation is required.

## Discussion

In our study, we have found that the GC content, length distribution, and randomness of eccDNA in rice are largely consistent with previous research[17,20,21]. This suggests that the formation of eccDNA is a widespread, random event across various species and tissue types.

However, our research has uncovered a unique phenomenon in rice: the leaves exhibit a high quantity of eccDNA. We hypothesize that under high light intensity, rice might augment the quantity of eccDNA to bolster its response to the reactive oxygen species (ROS) produced during photosynthesis. This adaptation could help maintain normal physiological functions in rice under intense light, reducing the risk of photodamage. As previously reported, the presence of eccDNA carrying herbicide resistance genes can confer herbicide resistance in plants, suggesting that eccDNA can influence the expression of specific genes[12,22,23]. We speculate that in rice leaves, the abundant eccDNA might be involved in regulating genes associated with photosynthesis and photoprotective mechanisms, aiding the plant's adaptation to intense light environments. Of course, this may not represent a direct regulation (Fig. 7a). Furthermore, this phenomenon reflects the evolutionary adaptability of rice to specific environmental stresses. Considering the formation of eccDNA, which may lead to chromosomal deletions and rearrangements[24], and in light of other literature citing

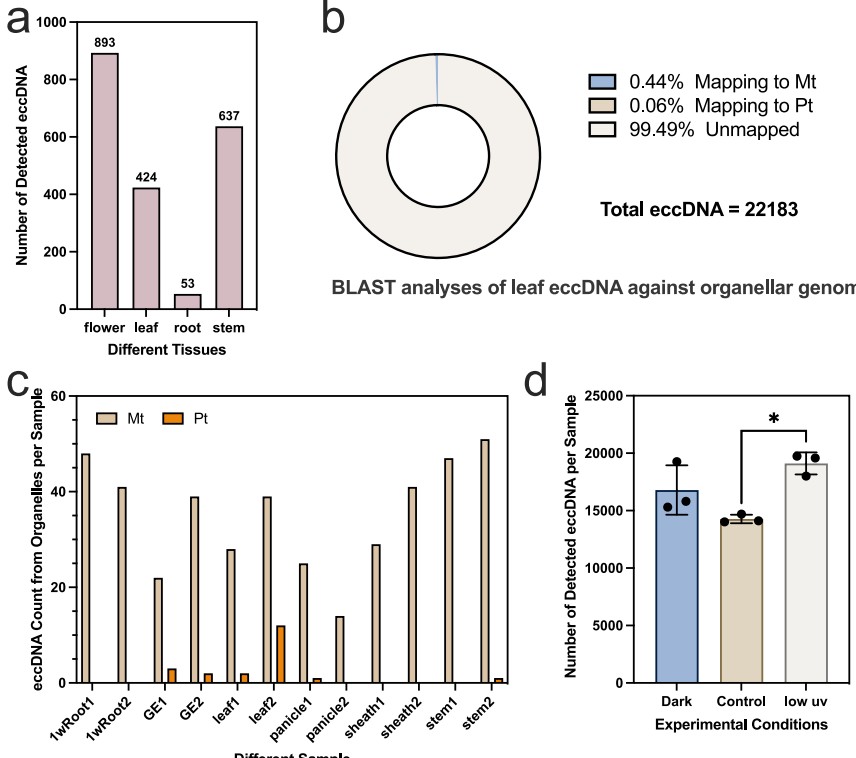

**Fig. 8 | Investigation of abundant eccDNA present in rice leaf tissues. a** All eccDNAs identified in four distinct tissues of *Arabidopsis thaliana*. **b** The results of BLAST analysis of leaf eccDNA against organellar genomes, with "Mt" representing the mitochondrial genome and "Pt" representing the chloroplast genome. **c** Examination of the presence of eccDNAs originating from mitochondria and chloroplasts across different rice tissues. **d** Quantification of eccDNA across various experimental conditions. This bar graph illustrates eccDNA molecules detected per sample, comparing Dark (no light exposure), Control (ambient light conditions), and Low UV (low ultraviolet light exposure) conditions. Analysis involved 3 pairs of samples, utilizing a two-tailed paired t test, which revealed a significant difference between Low UV and Control conditions ($p = 0.024$, df = 2). The error bars represent standard deviation (SD). Source data are provided as a Source Data file.

the role of eccDNA in facilitating 'dual-speed' genome evolution primarily by promoting instability at genomic hotspots[25], we hypothesize that rice leaves might utilize eccDNA to modify their genomic composition. This modification could enhance their tolerance to intense light, thereby conferring an evolutionary advantage. Finally, by comparing data from samples under drought stress, we find that only stress factors causing DNA damage led to an increase in eccDNA in rice. This indicates the potential of eccDNA as a practical biomarker for assessing DNA damage.

In 2012, Shibata, et al.[26] analyzed and compared microDNAs (200–400 bp eccDNAs) in mammalian cells, finding a significant proportion (37%) of these in mammals displayed microhomology. In our rice samples, although we detected a large fraction of eccDNAs, with about 50% falling within the 200–400 bp range, only approximately 7% of these eccDNAs exhibited sequences with such microhomology. This disparity may reflect fundamental differences in genome organization and stability between species, especially in plant species like rice.

Furthermore, we revealed that DR-eccDNAs' sequences are not confined to the 200–400 bp range but can include longer sequences, with the longest in our study reaching 4542 bp. Further analysis indicated that DR-eccDNAs are significantly longer than those without. This suggests that the presence of microhomology in eccDNAs is not limited by length, challenging previous study limitations[21,26,27], and implying that microhomology formation might be a more widespread genomic variation mechanism.

Importantly, we discovered that DR-eccDNAs and Other-eccDNAs appear to arise from distinct mechanisms. Given that direct repeats usually range from 7 to 15 bp, with some mismatches allowed at the ends, we hypothesize that DR-eccDNAs might be produced via an alternative end-joining (Alt-EJ) pathway during DNA repair, consistent

with recent studies[10]. In contrast, Other-eccDNAs may arise through various DNA repair pathways. Additionally, using TGS and PCR experiments, we found that eccDNA formation leads to the loss of chromosomal fragments, regardless of the presence or absence of associated microhomology, suggesting that eccDNA can induce chromosomal deletions. This deeper understanding, combined with previous research on the role of eccDNA in genomic rearrangements through circular chimeras and reintegration[24], and the involvement of reverse transcriptase transposons in eccDNA formation during the host's Alt-EJ DNA repair process[10], underscores the complex impact of eccDNA in driving genomic changes and its potential influence on genomic stability. Moreover, analysis of publicly available pan-genome data for rice reveals that structural variations in some rice varieties contain direct repeats, a characteristic feature of eccDNA[28]. Based on these facts, we propose that eccDNA serves as a catalyst for driving genomic variations in offspring. After all, once chromosome deletions induced by eccDNA occur in germ cells, they become heritable.

Previous studies have reported an abundance of eccDNAs associated with transposable elements (TEs)[29,30]. In plants, Lanciano, et al.[29] utilized the Mobilome-seq technique to identify a set of TE-related eccDNAs across multiple species. Interestingly, they discovered that PopRice, an active retrotransposon in rice, generates substantial eccDNAs in the endosperm of wild-type rice seeds. Subsequent research revealed that PopRice eccDNAs might play a regulatory role during the transition from seed to seedling in rice, particularly in responses related to plant hormones like gibberellins (GA) and abscisic acid (ABA)[31]. These findings offer a perspective on the role of LTR-retrotransposon-derived eccDNAs in plant development. However, these studies underscore the functional significance of eccDNAs in plant growth, primarily focusing on the relationship between TE

regions and eccDNAs. In fact, our study also found that eccDNA is significantly abundant in the TE region, especially in the LTR region. Aligning with the findings reported by Hu et al.[32], our research also uncovered a significant correlation between eccDNAs and DNA Repeats within TEs. This observation suggests the possibility that DNA transposons may undergo circularization, either as an alternative to or concurrently with reintegration into the genome, a hypothesis that warrants further exploration.

Additionally, our study suggests a potential link between eccDNAs and circRNAs. Contrary to previous reports, we did not observe a connection between eccDNAs and tRNAs[17,27]. Furthermore, we noted a significant enrichment of eccDNAs in the CDS regions of the genome, which does not directly affect gene expression[5,17,26,33]. Considering past research that suggests eccDNAs might indirectly influence gene expression through mechanisms like highly accessible chromatin and long-range gene regulation, promoting oncogene activation, or the non-coding regions in eccDNAs acting as potential enhancers, affecting the expression of oncogenes carried by eccDNAs[34,35]. These findings collectively indicate that eccDNAs are involved in a complex genomic regulatory network.

Overall, our research provides a crucial understanding of the genesis, distribution, and gene interactions of eccDNA, uncovering unexplored aspects of genomic architecture and necessitating further exploration of eccDNA's impact on chromosomal integrity. This substantial investigation exposes potential genomic disruptions introduced by eccDNA formation, the implications of which could be profound for organismal biology and herald crucial directions for future research. Leveraging this understanding, we posit the potential for developing new strategies to enhance crop adaptability and yield - a prospect of immense significance in the face of current environmental challenges. Essentially, our work underscores the potential of eccDNA studies to shape our comprehension of genomic integrity and its broader implications for agricultural advancements.

## Methods

### Materials

*Oryza sativa* L. ssp *japonica* cultivar Nipponbare was used for sample collection. For the extraction of eccDNA samples, our starting material was 100 mg of tissue powder from the Nipponbare cultivar. We prepared this tissue powder by grinding the samples under conditions of liquid nitrogen.

To obtain germinated embryo samples, we incubated rice seeds in water at 37 °C under dark conditions for three days. Subsequently, we carefully excised the radicles and coleoptiles using a razor blade. We harvested roots from one-week-old seedlings, which were grown in water, by employing scissors for the task. For the collection of stems, leaves, leaf sheaths, and panicles, we targeted rice plants at the pre-flowering stage. To compare control and drought-stress conditions, seedlings were incubated either in water (control) or a 20% PEG 6000 solution (drought treatment) for four hours. Following this, we excised the aboveground portions of these seedlings using scissors. All rice plants, except for those designated for germinated embryo samples, were cultivated under controlled environmental conditions: a 13-h light (29 °C) / 11-hour dark (26 °C) photoperiod.

To further our research, we conducted additional experiments with three replicates under varying conditions: darkness at 28 °C, a control setting, and UV exposure. The UV exposure condition was maintained at 29 °C with a 13-h light/11-h dark cycle, using 10-day-old samples from the Nipponbare cultivar. For the UV treatment specifically, we exposed the samples to a UV intensity of 78 μW/cm² for a duration of 20 minutes.

### Circle-seq

Circle-seq was conducted as described with several modifications[19]. Briefly, rice tissues were subjected to eccDNA extraction using the TIANprep Mini Plasmid Kit (TIANGEN). The crude extracts were incubated overnight with the rare-cutting restriction endonuclease *Mss*I (Thermo Scientific) to fragment residual chromosomal DNA and organellar DNA, which were further digested with the Plasmid-Safe ATP-Dependent DNase (Epicenter). The refinements were then amplified by REPLI-g Midi Kit (QIAGEN). The amplified products were ultrasonically sheared and converted into libraries for cluster generation and DNA sequencing. In this process, we utilized the circle-seq sequencing service provided by DIATRE Biotechnology (China).

### Paired-end sequencing for eccDNA identification

To identify eccDNA from Circle-seq data, we employed two independent methods. First, we used Circle-Map software (version 1.1.4) to identify junctions potentially formed after DNA cyclization[36]. We ran Circle-Map on BWA aligned reads, using default parameters, and filtered candidates as follows: (1) At least one discordant read and two independent split reads support the breakpoint coordinates; (2) Circle score, an additive scoring scheme that takes into account the alignment quality, length of the split segment, and the number of split reads supporting the circular DNA, greater than 50; (3) the coverage ratio at the start or end coordinate was set between 0.33-1, indicating high quality circles; and (4) the coverage continuity was set to 0, indicating that the entire circle was covered by reads. We generated a Potential eccDNA list for each sample and combined the results for further analysis. As a second approach, we used ecc_finder software (version 1.0.0) to identify Potential eccDNA in different samples (default parameters)[37]. To examine the support of these Potential eccDNA, we used bedtools (version 2.29.2)[38] to intersect two Potential eccDNA lists derived from separate methods, constructing a combined list of eccDNA coordinates. Lastly, taking into account a balance between the length and quantity of eccDNAs, we opted for an overlapping region threshold greater than 90%. A more detailed approach can be found in Supplementary Note 1 and Supplementary Fig. 6 and 7.

### Analysis of physical properties of eccDNA

Using the eccDNA list we constructed, we extracted the corresponding eccDNA coordinates and the sequence of each 150 bp region before and after to calculate the GC content. We used Circle-Map software to evaluate and score eccDNA, recorded as the Circle score. The Circle score is a scoring scheme that takes into account the alignment quality, the length of the split segments, and the total number of split reads. Drawing inspiration from the methodology described by Prada-Luengo et al.[39] in the existing literature, we implemented a 5% incremental subsampling technique, building upon our own established eccDNA identification method. This approach allowed us to perform a comprehensive saturation analysis on all samples (Supplementary Note 1; Supplementary Fig. 8).

### Direct and inverted repeat analysis of eccDNA

To analyze the sequence characteristics of eccDNA, we extended the end coordinates of eccDNA by 100 bp outward and contracted them by 50 bp inward and used short blastn to identify inverted and direct repeat sequences in the resulting sequence[40]. To evaluate the reliability of the identified eccDNA, we constructed a random model based on the length distribution of eccDNA. The model contains 6000 randomly generated DNA sequences with random lengths and positions. For specific parameter settings, see the script (eccDNA_Repeat_Classifier.py) provided by us.

### PCR validation of eccDNAs

The PCR assay was performed with 2 × Taq Plus Master Mix (Vazyme). The PCR products were analyzed by electrophoresis on 3% agarose gel. All the primers used in experiments can be obtained in the Supplementary Data 8. *DNA exonuclease assay*: eccDNA crude extracts were mixed with the Exonuclease V (New England Biolabs). The prepared

reaction was divided equally into two parts. One was kept on ice and served as control, the other incubated at 37 °C for 30 min. After that, in both reactions, EDTA was added to 11 mM, followed by heat inactivation at 70 °C for 30 min. The treated samples were then used directly as PCR templates with Phanta Max Super-Fidelity DNA Polymerase (Vazyme). After checked on 3% agarose gel, the PCR products were cloned with *pEASY*-Blunt Cloning Kit (TransGen) and the resultant clones were sequenced. *Recapitulation of eccDNA formation:* The full-length eccDNA sequences with or without the flanking direct repeat sequences were amplified with Phanta Max Super-Fidelity DNA Polymerase (Vazyme) and cloned with *pEASY*-Blunt Cloning Kit (TransGen). Clones with inserts in both orientations were selected and multiplied for plasmids preparation. The purified plasmids were directly used as PCR templates with 2 × Taq Plus Master Mix (Vazyme). The PCR products were analyzed by electrophoresis on 3% agarose gel. *Chromosomal microdeletion detection:* PCR was performed with Phanta Max Super-Fidelity DNA Polymerase (Vazyme) without extension steps to avoid amplification of the much abundant intact counterpart. The PCR products of the deleted template were cloning with *pEASY*-Blunt Cloning Kit (TransGen) and the resultant clones were sequenced. *Nested PCR:* In the first round of PCR, eccDNA crude extracts were used as templates. After electrophoresis, 1.5 μL of the PCR products were added in a 15 μL reaction of the second round of PCR. Both rounds of the PCR were performed with Phanta Max Super-Fidelity DNA Polymerase (Vazyme). The second round PCR product from young leaf sample was cloning with *pEASY*-Blunt Cloning Kit (TransGen) and the resultant clones were sequenced.

## Genome annotation of eccDNA

We used the intersect function of bedtools to annotate eccDNA based on the MSU v7 Rice reference genome[41]. The positions of genes, repeat units and eccDNA on chromosomes are displayed by TBtools[42] and Circos[43]. For each eccDNA, we measured its overlap with various annotated features in the genome (e.g., intergenic region, genic region, 5′UTR, CDS, intron, 3′UTR, repeat units, miRNA, tRNA). To be included in the analysis, eccDNA must have at least 50% overlap with the annotated feature. To evaluate whether eccDNA is enriched in specific regions of the genome, we compared the locations of detected eccDNA with those of the stochastic model. We used paired t test to determine statistical significance. In addition, all the GO enrichment analyses of eccDNA overlapping genes were done with the help of clusterProfiler package and R. The minimum gene set was 5, the maximum gene set was 5000, and $p$ value < 0.05. The All eccDNA overlapped Gene are all list in Supplementary Data 9.

## RNA sequencing

RNA-seq data was analyzed using fastp (version 0.22.0) to trim adaptors and low-quality reads[44]. The cleaned paired-end reads were aligned to the MSU v7 Rice reference genome using Hisat2 (version 2.2.1)[45]. Samtools (version 1.3.1) was used to process the aligned sam file, producing a sorted bam file[46]. FeatureCounts (version 2.0.1) was used to quantify gene expression levels in the bam file using the reads aligned to the transcriptome as input[47]. The default parameters were used by all the software listed above. Differentially expressed genes between control and drought-treated seedlings were identified using edgeR packages[48] with the following rules: |Fold Change| >1 and p value < 0.05. All RNA-seq data can be obtained from Supplementary Data 10.

## Genome sequencing

As shown in Fig. 5a, our study adopted an experimental approach that integrated NGS and TGS technologies to investigate the presence of chromosomal deletions caused by eccDNA in three distinct rice samples. Firstly, we employed the PacBio Sequel IIe sequencing platform for third-generation high-fidelity (HiFi) sequencing of 'Nipponbare'

rice. We began by extracting high molecular weight DNA from fresh leaf tissues of healthy Nipponbare plants, followed by stringent quantification and quality assessments. The DNA library was then constructed using the PacBio SMRTbell Template Prep Kit 1.0, followed by HiFi sequencing on the Sequel IIe platform. Post-sequencing, the data were processed using PacBio's SMRT Link software, which included filtering, adapter sequence removal, and generating HiFi reads. This process yielded high-quality TGS genome data of Nipponbare. For the two NGS samples, the Nip8 sample was directly obtained from the previous study by Zhao et al.[49], while the Nip5 sample was prepared by us following the same experimental procedure. Then, we began by using the bedtools flank tool to obtain sequences 200 base pairs upstream and downstream of target genes, concatenating them to form a 400 base pair fusion sequence. A blacklist database was then constructed by BLAST-aligning these synthesized fusion sequences with the reference genome to identify and exclude potential background noise. Subsequently, two NGS datasets and one TGS dataset were converted to fasta format and BLAST-aligned against the fusion sequence database, aiding in the detection and validation of potential eccDNA events. To identify eccDNA of biological significance, we filtered the BLAST results using the awk tool, retaining only sequences that met specific alignment quality thresholds. Ultimately, we employed Venn diagrams to filter and identify genuine deletions caused by eccDNA, effectively distinguishing them from false positives.

## Reporting summary

Further information on research design is available in the Nature Portfolio Reporting Summary linked to this article.

## Data availability

The sequencing datasets generated in this study have been deposited in the European Bioinformatics Institute (EBI) database under accession PRJEB59090. In the context of project, the collected samples can be broadly classified into three distinct categories: Circle-seq, RNA-seq, and genomic sequencing samples. The Circle-seq dataset, spanning samples ERR10889836 to ERR10889855, encompasses a diverse array of biological specimens. This collection includes seedlings subjected to drought stress, embryos in the germination phase, newly developed leaves undergoing specific treatments, seedlings from the control group, panicles of rice, roots aged one week, as well as leaf sheaths and stems. Parallelly, the RNA-seq dataset (samples ERR10889820 to ERR10889835) provides a comprehensive transcriptional profile of these varied tissues, offering insights into their gene expression dynamics. Complementing these datasets, the project also includes three pivotal genome resequencing samples: these consist of the Next-Generation Sequencing (NGS) whole-genome sequence of the rice variety Nipponbare (ERR11838732), its HiFi Whole Genome Sequencing (WGS) counterpart (ERR11838731), and an additional Nipponbare sample from project PRJEB19404 (ERR2245546), denoted in this study as "Nip8". These samples play a crucial role in facilitating a nuanced analysis of genomic variations within this species. Source data are provided with this paper.

## Code availability

scripts used in this article are available in the GitHub repository [https://github.com/YxZhang-XHCY/eccToolkit].

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

## Acknowledgements

This study received financial support from the National Natural Science Foundation of China, awarded to T.L. and B.H. under grant numbers 31370025 and 31788103. We extend our heartfelt thanks to Prof. Xuehui Huang, Prof. Lin Xu, Prof. Xinguang Zhu, and Dr. Xiaoxiang Ni for their invaluable advice and mentorship throughout this research. Our appreciation also goes to BioRender, whose tools and resources were instrumental in enhancing the visual representation of our data, significantly improving the clarity and impact of our findings. Additionally, we are grateful for the analytical and graphical capabilities provided by GraphPad Prism version 10.2.0 for Mac (GraphPad Software, Boston, Massachusetts, USA, www.graphpad.com), which played a pivotal role in our data analysis and presentation.

## Author contributions

T.T.L. designed the study, performed computational analyses and wrote the manuscript. B.H. co-designed the study and polished the manuscript. J.D.Z. carried out collection of all samples and other experiments except sequencing and participated in writing the manuscript. Y.X.Z. performed all computational analyses and participated in writing the manuscript. C.C.Z., D.L.F., Q.F. and Y.Q.L. performed sequencing. T.H. gave technical supports on IT. Y.Z and Q.Z. gave advice for this study. All authors read and approved the final manuscript.

## Competing interests

The authors declare no competing interests.
