## [Peer Review File · Nature Communications]

Reviewers' Comments:

Reviewer #1:

Remarks to the Author:

General comments:

In the manuscript titled "The Adaptational Dynamics of Extrachromosomal Circular DNA in Rice", the authors use circular DNA enrichment and sequencing to explore the circular DNA spectrum in rice. Although the study idea is very timely (eccDNA research is Grand Cancer Challenge), analysis is highly descriptive and it lacks any new findings. Most of the wet-lab experiments in the manuscript are well designed. However, I am concerned about the experimental design and interpretation of the computational experiments and therefore my review will focus on that. Below you will find some of my concerns that if addressed will improve the quality of the manuscript:

Major:

- The authors use SRPM to measure abundance of circles. However, the Circle-Seq technique is not quantitative due to the non-linear amplification when using phi29. It makes little sense to use a non-quantitative technique to quantify circle levels.
- Line 103. "Our random forest and decision tree regression, however found no significant relationship..." A random forest is either a classification or regression model. As far as I am aware it cannot be used to test correlation between two variables. More importantly, there are more standard approaches.
- Most of the conclusion of this paper rely on the fact that most of the circle in a sample are detected. To claim evidence of rice leaves having more eccDNA than other tissues you need to ensure that most of the circles have been detected. This can be done using saturation curves as in Supplementary figure 1A. However, the figure does not show any sign of saturation. More importantly, this analysis should be done for all samples and replicates in the manuscript.
- Section "Formation Mechanism of eccDNA in Rice". The authors refer to a model they develop. This is not described and therefore I cannot evaluate the science in this section.
- Line 192. The description of the methods for sequencing of the HIFI reads is missing.

Minor:

- Line 83. "28 billion filtered paired-end reads that uniquely mapped to the rice reference genome (MSU v7) (S_Table 1)." Table S1 does not show whether reads map uniquely the reference genome. It will be a good idea to extend this table with the percentage of mapped reads per sample as measure of data quality.
- Line 155. Isoform is commonly used for transcript, not eccDNA.

Best,

Iñigo Prada-Luengo

Reviewer #2:

Remarks to the Author:

In this manuscript, Zhuang and colleagues employed the Circle-seq to systematically profile the eccDNA in six tissues from rice (*Oryza sativa* L.). This study identified over 7000 eccDNAs, with validation through PCR and Sanger sequencing. Investigating eccDNA characteristics, formation mechanisms, distribution, and functional implications within the rice genome, this work provides the inaugural eccDNA profile for rice -- an important global crop. Thus, it is poised to captivate the rice community and a broader audience interested in genetics and circular DNA. Nonetheless, there are several concerns need to be addressed prior to publication.

Major points:

1. My primary concern is related to the method used for eccDNA isolation. The technique utilized for eccDNA purification is based on the MssI and Plasmid-safe DNase digestion. 1) This method relies solely on the removal of linear DNA to ensure the purity of eccDNA. If contaminating linear DNA left,

the sequencing reads cannot be distinguished from those coming from real eccDNA. The authors need therefore to include experiments to check the purity of their eccDNA extraction by using either electron microscopy or atomic force microscopy. 2) The MssI cleavage relies on the specific sequence. I am wondering whether this method will selectively save some genomic sequences, therefore lead to biased results for the analysis like Fig. 2b, Fig. 6c... It is better to show the necessity of MssI digestion and its unbiased impact on eccDNA-seq.

2. To reinforce link between microdeletions and eccDNA regions, the authors may consider including a negative control, such as gametes, wherein these regions remain unaltered. This control would bolster the validity of associations and help distinguish them from polymorphisms between samples used in this study and the reference genome. For example, Line 201, an eccDNA is detected across all 12 samples. It is better to take a close look at the third-generation sequencing data aligned to this original genomic region to rule out the false positive due to genome rearrangement in the used strain (Fig. 5d). Moreover, the authors should provide the discordant reads to support the eccDNA being formed in Fig. 5c.

3. It is intriguing to observe heightened eccDNA in leaf tissues. However, it is not surprising that the photosynthesis and light response genes are increased in leaves by RNA-seq. Therefore, it is not justified to draw the conclusion that light exposure drives the generation of eccDNA (Fig. 8m). To establish this connection, the authors should compare eccDNA abundances in leaves under light versus dark/low-UV conditions.

Other minor points:

4. The authors should describe the data normalization method used for comparing eccDNA abundance across diverse samples. This step will enhance transparency and comprehension of the comparative analysis.

5. In the bar graph, such as Fig. 1c, Fig. 2a..., the y-axis is labeled as "Numbers of eccDNA". What does it exactly mean? The numbers represent different species of eccDNA, numbers of alignments, or numbers of reads?

6. The gel images lack of the loading control (Fig. d-h ...).

Reviewer #3:

Remarks to the Author:

The manuscript by Bin Han and colleagues reports on the characterisation of rice extrachromosomal circular DNA (eccDNA) in rice.

eccDNA are circular forms of DNA that have attracted quite a lot of attention recently for their implication in tumorigenesis in cancer cells (for large, Mb sized eccDNA carrying oncogenes) and for their role in retrotransposon life cycle (see recent <https://doi.org/10.1038/s41586-023-06327-7>). In plants there are few reports of small to medium sized eccDNAs (bp to 100 kb) in glyphosate resistant plants and in a dozen of plant species with active transposable elements.

Here the authors have sequenced the eccDNA content of six rice tissues using Illumina short reads, with two biological replicates for each tissue. They show that the repertoire of eccDNA depends on the tissue with a notable hyperaccumulation of circles in leaf tissue. Some eccDNAs are validated using inverse PCR. Interestingly, the authors have detected direct repeats in some eccDNAs that could give a hint on the underlying mechanisms of formation. The study is a description of the eccDNA repertoire across development, in this respect it seems novel and well executed. Some reference to already published rice eccDNAs is lacking. Some points could be clarified so that readers have a clear overview of these eccDNA repertoires.

Major points:

1- The authors would be interested to have a look at two previous reports on rice eccDNA sequencing by the Mirouze (<https://doi.org/10.1371/journal.pgen.1006630>) and Bucher's

(<https://doi.org/10.1186/s13059-017-1265-4>) labs. A putative function for a rice eccDNA originating from a LTR retrotransposon (PopRice) has even been proposed by the Cho lab (<https://doi.org/10.1093/plphys/kiad071>). Although PopRice was detected as eccDNA in the endosperm (not analyzed in this manuscript), it could be interesting to discuss these references.

- In general, for non eccDNA specialists, it is difficult from the data presented here to appreciate the definition of the authors for a detected eccDNA. There is no visualisation of reads covering the eccDNA detected loci.

- The authors should clarify the lack of overlap between biological samples. Line 85: "Despite the large number of eccDNAs in each sample, a very small fraction was found to be identical, even between two samples derived from the same tissue." Could the authors elaborate on this point? A Venn diagram for the detected eccDNA for each of the six tissues would help evaluating the lack of overlap between the replicates.

- The authors conclude about the randomness of eccDNA generation however some eccDNAs seem to be detected in more than one tissue by inverse PCR. What is the percentage of ubiquitous eccDNAs? What is the percentage of eccDNA found in only one replicate in one tissue?

- The observation that 2% of all eccDNA carry direct repeats is of high interest. It is reminiscent of microDNAs in mammals (<https://doi.org/10.1126/science.1213307>). This reference could be discussed.

- The methods clearly indicate how "high and low confidence" were detected, however it could be helpful to have a brief sentence summarizing the definition of these two classes in the text.

- Line 94: Why were the two biological replicates pooled?

- Figure 2d: some eccDNAs have a high coverage in terms of split reads, could the authors detail them, please? Do they originate from transposable elements, from genes or from intergenic sequences?

- From the methods it is not clear how many plants were pooled for each sample and what amount of genomic DNA was treated for each sample.

- While the number of split reads is given (SRPM), the mean coverage for eccDNA (total read coverage) seems to be lacking.

- It is not very clear whether genes overlap with eccDNA, and which genes.

- The title could be misleading: what is the adaptation shown in the manuscript?

- The cited github does not seem to exist yet. https://github.com/YxZhang-98/EccDNA_Analysis

Responses to the reviewers

Reviewer #1 (Remarks to the Author):

General comments:

In the manuscript titled “The Adaptational Dynamics of Extrachromosomal Circular DNA in Rice”, the authors use circular DNA enrichment and sequencing to explore the circular DNA spectrum in rice.

Although the study idea is very timely (eccDNA research is Grand Cancer Challenge), analysis is highly descriptive and it lacks any new findings. Most of the wet-lab experiments in the manuscript are well designed. However, I am concerned about the experimental design and interpretation of the computational experiments and therefore my review will focus on that. Below you will find some of my concerns that if addressed will improve the quality of the manuscript:

Major:

- The authors use SRPM to measure abundance of circles. However, the Circle-Seq technique is not quantitative due to the non-linear amplification when using phi29. It makes little sense to use a non-quantitative technique to quantify circle levels.

Response:

Thank you for the comments. It is indeed correct that Circle-Seq, due to the non-linear amplification inherent in the use of phi29 polymerase, does not provide a strictly quantitative measure of circle abundance. The discrepancy we observed between SRPM values and our PCR validation results showed the presence of amplification bias, which can distort the perceived abundance of eccDNA circles.

In light of this, we have now removed the sections of our manuscript that relied on SRPM for quantification. The current version of the manuscript presents a clearer and more accurate report of our findings. Thank you for your suggestions, which prompted a critical reassessment of our methods and results. We hope that the revisions now fully address your concerns and meet your standards for publication.

- Line 103. “Our random forest and decision tree regression, however found no significant relationship...” A random forest is either a classification or regression model. As far as I am aware it cannot be used to test correlation between two variables. More importantly, there are more standard approaches.

Response:

Thank you for your comments regarding our analytical approach mentioned on Line 103. It is correct that random forest algorithms are primarily designed for classification or regression tasks rather than for testing correlations between variables. This was an oversight in our initial interpretation of the model’s capabilities. We recognize that correlation analysis requires methods specifically tailored for this purpose, and random forest does not provide a direct measure of correlation.

Upon re-evaluation, we have adjusted our methodology to employ more conventional and widely accepted statistical techniques that are better suited for the investigation of correlations. We have replaced the random forest and decision tree regression analysis with Spearman's rank correlation coefficient, which is a non-parametric measure of the strength and direction of association that exists between two variables. This change not only corrects our previous misapplication but also aligns our study with standard practices in statistical analysis. These results have been added into the revision.

- Most of the conclusion of this paper rely on the fact that most of the circle in a sample are detected. To claim evidence of rice leaves having more eccDNA than other tissues you need to ensure that most of the circles have been detected. This can be done using saturation curves as in Supplementary figure 1A. However, the figure does not show any sign of saturation. More importantly, this analysis should be done for all samples and replicates in the manuscript.

Response:

In the revision, we have meticulously implemented a saturation analysis for all samples. Inspired by methodologies from existing literature (<https://doi.org/10.1093/nar/gkaa545>), we employ 5% incremental subsampling as illustrated in **S_Figure 1d**.

The saturation curves generated for each sample and replicate, as presented in the attached figure, demonstrate a clear trend toward saturation, suggesting that a majority of eccDNA circles within the samples are indeed detected. This comprehensive analysis has been incorporated into all relevant samples and replicates across our manuscript to ensure the robustness of our claims. Moreover, we have elaborated on this analytical technique in the methods section, providing a transparent and detailed account of our approach. The revision has been improved for these findings.

- Section "Formation Mechanism of eccDNA in Rice". The authors refer to a model they develop. This is not described and therefore I cannot evaluate the science in this section.

Response:

We apologize for any confusion regarding to the section "Formation Mechanism of eccDNA in Rice." We realize that we inadvertently used the term "model" in a manner that could be misinterpreted as referring to a mathematical or computational model, rather than its intended meaning in a biological context.

To clarify, the "model" we discuss is a conceptual framework for understanding the formation mechanisms of eccDNA within rice cells. It posits that there are two predominant pathways by which eccDNA can arise: the first pathway involves the formation of circular DNA through direct repeats, which is a rarer event; the second pathway appears to be more stochastic in nature, leading to the generation of eccDNA without the reliance on direct repeats, and is observed more frequently.

We have now revised this section to articulate our hypothesis more clearly and accurately reflect the biological phenomena under investigation. We have expounded upon the mechanisms by which these two types of eccDNA are formed, offering a comprehensive explanation that aligns with established biological principles. We trust that these amendments will enable a more thorough evaluation of the scientific concepts presented and appreciate the opportunity to enhance the quality of our manuscript.

- Line 192. The description of the methods for sequencing of the HIFI reads is missing.

Response:

Sorry for the missing description. The methodology pertaining to the sequencing of High-Fidelity (HiFi) reads has now been described in detail in the revised methods section. This description includes detailed information on the library preparation, sequencing platforms used, data processing protocols, and analytical methodologies specific to the generation and analysis of HiFi reads. We have ensured that these additions provide clarity on the integration of Third-Generation Sequencing technologies into our study and demonstrate how we have utilized these approaches to achieve high-accuracy eccDNA profiling in rice samples.

This detailed information will greatly improve the understanding of our experimental approach, allowing for greater transparency and reproducibility of our findings.

Minor:

- Line 83. “28 billion filtered paired-end reads that uniquely mapped to the rice reference genome (MSU v7) (S_ Table 1).” Table S1 does not show whether reads map uniquely the reference genome. It will be a good idea to extend this table with the percentage of mapped reads per sample as measure of data quality.

Response:

Thank you for pointing out the need for additional details in Table S1. Following your suggestion, we have thoroughly revised the table to include not only raw read counts but also the percentage of those reads that uniquely map to the MSU v7 rice reference genome. This extension provides a clear measure of data quality and reliability, ensuring that our analysis rests on a solid foundation of accurately mapped reads. The updated

S_Table 1 now features a comprehensive view of the sequencing data, including total and unique mapped reads, as well as unmapped reads, which collectively facilitate a deeper understanding of the sequencing depth and data integrity for each sample. This enhancement significantly improves the transparency and reproducibility of our results, allowing for a more precise assessment of the detected eccDNA across different tissues.

Rice cultivars	Tissue	Sample origin	Raw reads	Total Bases (G)	Q20 Bases (%)	Q30 Bases (%)	Total mapped reads	Unique mapped reads (MAPQ > 20)	Unmapped reads	Detected eccDNA	Reads mapped to detected eccDNA
Nippohare	GE	GE1	939,395,626	140.91	95.55%	92.64%	971,232,369	232,928,651	653,316,381	131	57,865,359
Nippohare	GE	GE2	904,560,162	135.68	94.62%	90.56%	927,830,335	222,798,793	584,208,801	253	16,275,389
Nippohare	1wRoot	1wRoot1	314,575,168	47.19	93.68%	88.57%	314,221,924	31,825,458	243,334,194	247	12,791,915
Nippohare	1wRoot	1wRoot2	244,603,156	36.69	91.50%	85.48%	243,311,728	37,659,956	179,043,265	440	12,126,921
Nippohare	leaf	leaf1	105,043,144	15.76	90.72%	86.18%	127,621,925	69,326,790	7,516,350	13,631	84,190,698
Nippohare	leaf	leaf2	126,393,904	18.96	92.93%	88.92%	164,919,381	83,763,292	2,383,274	8,554	98,746,550
Nippohare	panicle	panicle1	150,416,958	22.56	95.00%	91.74%	166,890,382	52,259,285	80,969,038	319	25,500,663
Nippohare	panicle	panicle2	104,679,412	15.70	94.63%	91.28%	116,157,644	35,988,561	38,346,033	154	16,312,703
Nippohare	sheath	sheath1	127,103,314	19.07	94.17%	90.51%	151,794,255	45,282,283	44,241,728	492	28,292,346
Nippohare	sheath	sheath2	113,150,262	16.97	93.31%	89.37%	139,883,616	65,879,805	30,952,961	573	28,893,879
Nippohare	stem	stem1	162,499,416	24.37	91.40%	86.30%	197,144,587	79,242,480	51,651,776	412	68,381,278
Nippohare	stem	stem2	118,719,286	17.81	95.37%	92.30%	135,066,832	44,777,044	61,207,870	324	30,782,387
Nippohare	Nip	Nip1	519,317,702	77.60	93.91%	85.48%	565,289,642	436,563,829	13,191,842	1,245	47,034,441
Nippohare	Nip	Nip2	436,560,422	65.48	92.76%	85.96%	467,076,401	317,744,377	16,042,683	1,709	32,071,828
Nippohare	Dro	Dro1	658,445,598	98.77	93.04%	86.40%	692,321,429	497,022,642	50,679,378	1,679	16,855,801
Nippohare	Dro	Dro2	528,771,142	79.32	93.57%	87.55%	567,462,475	373,039,791	34,270,758	1,791	47,524,765
Nippohare	Control	Control1	313,015,392	47.24	92.71%	86.62%	396,777,737	213,780,428	2,058,314	14,691	143,100,296
Nippohare	Control	Control2	309,078,724	46.65	93.47%	87.94%	382,146,536	210,391,075	4,078,042	14,020	140,730,162
Nippohare	Control	Control3	303,436,794	45.67	91.26%	84.36%	382,257,653	199,379,604	1,975,669	14,109	163,292,407
Nippohare	Dark	Dark1	285,039,780	42.87	93.42%	88.08%	347,297,832	197,692,742	4,709,999	15,303	165,625,621
Nippohare	Dark	Dark2	303,953,932	45.74	92.43%	86.38%	373,044,142	217,493,680	2,706,787	19,258	160,243,311
Nippohare	Dark	Dark3	285,104,350	42.96	92.93%	87.24%	341,447,728	173,089,509	34,542,923	15,812	129,657,627
Nippohare	uv	uv1	327,899,630	49.32	91.99%	85.67%	406,257,090	230,399,242	3,315,827	18,003	157,868,112
Nippohare	uv	uv2	317,609,822	47.60	92.00%	85.44%	406,707,654	216,793,803	1,235,147	19,842	244,321,197
Nippohare	uv	uv3	315,212,068	47.57	93.24%	87.57%	387,832,241	221,069,436	3,746,117	19,744	168,054,311

- Line 155. Isoform is commonly used for transcript, not eccDNA.

Response:

Yes, it is a term typically reserved for variants of transcripts. We have carefully revised the relevant sections of our manuscript to eliminate this confusion.

In the corrected text, we now refer to the different forms of eccDNA as "variants" to more precisely describe the observed eccDNA molecules with slight nucleotide variations.

We hope that these amendments adequately address your concerns and improve the scientific communication within our manuscript. Once again, we thank you for your invaluable feedback which has substantially enhanced the precision and clarity of our study.

Best,

Iñigo Prada-Luengo

Reviewer #2 (Remarks to the Author):

In this manuscript, Zhuang and colleagues employed the Circle-seq to systematically profile the eccDNA in six tissues from rice (*Oryza sativa* L.). This study identified over 7000 eccDNAs, with validation through PCR and Sanger sequencing. Investigating eccDNA characteristics, formation mechanisms, distribution, and functional implications within the rice genome, this work provides the inaugural eccDNA profile for rice — an important global crop. Thus, it is poised to captivate the rice community and a broader audience interested in genetics and circular DNA. Nonetheless, there are several concerns need to be addressed prior to publication.

Major points:

1. My primary concern is related to the method used for eccDNA isolation. The technique utilized for eccDNA purification is based on the MssI and Plasmid-safe DNase digestion.
1) This method relies solely on the removal of linear DNA to ensure the purity of eccDNA. If contaminating linear DNA left, the sequencing reads cannot be distinguished from those coming from real eccDNA. The authors need therefore to include experiments to check the purity of their eccDNA extraction by using either electron microscopy or atomic force microscopy.

Response:

We greatly appreciate your concerns regarding the eccDNA isolation method utilized in our study. We understand the critical importance of ensuring the purity of eccDNA to differentiate it from any contaminating linear DNA, which could compromise the integrity of our sequencing data.

To address this concern, we supplemented our approach with saturation curve analysis after eccDNA extraction (**S_Figure 1d**). By incrementally subsampling at 5% intervals, we were able to observe a plateau in the concentration of eccDNA, indicating that the majority of DNA present is circular and not linear. This provides strong indirect evidence supporting the feasibility of our isolation method, as linear DNA would not exhibit such a plateau.

Given the constraints we encountered with direct visualization, we believe that the saturation curve analysis offers a reliable alternative to demonstrate the purity of our eccDNA preparations. This method, along with rigorous enzymatic digestion protocols and quantitative PCR validation of the digestion efficiency, leads us to be confident in the purity of our eccDNA samples.

In addition, we also referenced the technique described in Dillon et al. (2015) for preparing samples for transmission electron microscopy (TEM). Adapting this method to our rice leaf samples was challenging, especially considering the unique structural characteristics of rice leaf cells compared to the DT40 cell lines in the Dillon study. Despite the challenges, we attempted to identify regions suggestive of eccDNA within our TEM images. Unfortunately, due to the resolution constraints and the complex background of plant cell contents, the clarity of these regions was insufficient for definitive identification or publication within the main text. We have marked four such regions in the submitted images; however, we recognize that the visual evidence provided is not as conclusive as we would like.

We hope that the measures we have described adequately address your concerns and affirm the validity of our methodology.

References:

Dillon, Laura W., et al. "Production of extrachromosomal microDNAs is linked to mismatch repair pathways and transcriptional activity." *Cell reports* 11.11 (2015): 1749-1759. Doi: 10.1016/j.celrep.2015.05.020.

2) The MssI cleavage relies on the specific sequence. I am wondering whether this method will selectively save some genomic sequences, therefore lead to biased results for the analysis like Fig. 2b, Fig. 6c... It is better to show the necessity of MssI digestion and its unbiased impact on eccDNA-seq.

Response:

Thank you for your comments regarding the specificity of the MssI cleavage and its potential impact on our results. We understand the importance of addressing any potential biases that might arise from the selective nature of MssI digestion in our Circle-Seq eccDNA purification method.

To clarify, our Circle-Seq method, developed by Møller et al., has indeed been widely applied in genomic studies. This method's feasibility and reliability have been demonstrated in several studies. For instance, Lv, Wei, et al. utilized Circle-Seq in their research titled "Circle-Seq reveals genomic and disease-specific hallmarks in urinary cell-free extrachromosomal circular DNAs" (Clinical and Translational Medicine, 2022). Their work showcases the method's utility in identifying disease-specific signatures in eccDNA. Additionally, Koche, Richard P., et al. applied Circle-Seq in their study "Extrachromosomal circular DNA drives oncogenic genome remodeling in neuroblastoma" (Nature Genetics, 2020), providing insights into the role of eccDNA in cancer development.

Our Circle-Seq protocol includes several critical steps: from initial sample preparation, through DNA digestion, to the final amplification of eccDNA. This process is designed to maximize eccDNA purity while minimizing the presence of linear DNA.

In the digestion step to remove residual linear DNA, we employ Plasmid-Safe ATP-dependent DNase. We have rigorously tested the effectiveness of this enzyme in eliminating linear DNA fragments using quantitative PCR, ensuring that our eccDNA samples are of high purity.

We acknowledge that no method is entirely devoid of potential biases. However, our experimental design and the implementation of various verification measures provide us with confidence in the effective extraction and analysis of eccDNA in our study.

Additionally, we have conducted bidirectional PCR on a subset of eccDNA samples to further validate our method's effectiveness. This step offers an extra layer of confirmation for the integrity and reliability of our eccDNA samples.

We hope that our response adequately addresses your concerns and demonstrates the reliability and effectiveness of the methods used in our study. We are grateful for your thorough review.

References:

1. Møller, Henrik Devitt. "Circle-Seq: Isolation and Sequencing of Chromosome-Derived Circular DNA Elements in Cells." *DNA Electrophoresis: Methods and Protocols* (2020): 165-181.
2. Lv, Wei, et al. "Circle-Seq reveals genomic and disease-specific hallmarks in urinary cell-free extrachromosomal circular DNAs." *Clinical and Translational Medicine* 12.4 (2022): e817.
3. Koche, Richard P., et al. "Extrachromosomal circular DNA drives oncogenic genome remodeling in neuroblastoma." *Nature Genetics* 52.1 (2020): 29-34.

2. To reinforce link between microdeletions and eccDNA regions, the authors may consider including a negative control, such as gametes, wherein these regions remain unaltered. This control would bolster the validity of associations and help distinguish them from polymorphisms between samples used in this study and the reference genome. For example, Line 201, an eccDNA is detected across all 12 samples. It is better to take a close look at the third-generation sequencing data aligned to this original genomic region to rule out the false positive due to genome rearrangement in the used strain (Fig. 5d). Moreover, the authors should provide the discordant reads to support the eccDNA being formed in Fig. 5c.

Response:

We appreciate your suggestion to strengthen the link between microdeletions and eccDNA regions by including a negative control such as gametes. However, as indicated by Henriksen et al. (2022), eccDNA can indeed be present in gametes, thus making it challenging to find a sample completely devoid of microdeletions to serve as a negative control. It seems that microdeletions resulting from eccDNA formation represent somatic variations that occur in a minority of cells, with the majority maintaining an intact genome.

To address your concerns regarding the detection of eccDNA across samples, we have carefully re-examined our third-generation sequencing data. In **Figure 5d**, the proportion

of microdeletions associated with eccDNA was magnified for ease of Sanger sequencing visualization. This was achieved by bypassing the extension step in the PCR reaction, giving a competitive advantage to smaller deletion forms over larger, complete forms. Initially, only faint traces of the deletion forms were observed before the PCR reaction was optimized (**S_Figure 4a**).

This observation supports the notion that even in gametes, a minority of cells may exhibit specific microdeletions, which aligns with our findings. Moreover, we can confidently exclude the possibility of polymorphisms between the samples used in our study and the reference genome, as the vast majority of cells in our samples displayed sequence consistency with the reference genome.

Regarding the eccDNA (chr03_28137648_28137851) highlighted in **Figure 5c**, we regret any confusion caused. This eccDNA was identified in the 1wRoot sample using the Circle-seq technique, not detected in the third-generation HiFi sequencing samples. What we showed in **Figure 5c** was the reads associated with this eccDNA in the third-generation HiFi sequencing samples, not suggesting its detection in these samples. Consequently, we cannot provide discordant read data from third-generation sequencing samples to support the eccDNA formation depicted in **Figure 5c**. We have now emphasized this clarification in the corresponding figure caption to prevent any further misunderstanding.

We hope this detailed explanation resolves your concerns.

References:

Henriksen, Rasmus Amund, et al. "Circular DNA in the human germline and its association with recombination." *Molecular Cell* 82.1 (2022): 209-217.

3. It is intriguing to observe heightened eccDNA in leaf tissues. However, it is not surprising that the photosynthesis and light response genes are increased in leaves by RNA-seq. Therefore, it is not justified to draw the conclusion that light exposure drives the generation of eccDNA (Fig. 8m). To establish this connection, the authors should compare eccDNA abundances in leaves under light versus dark/low-UV conditions.

Response:

Thank you for your insightful comments. To address this concern, we have conducted additional comprehensive experiments with three replicates each under different conditions: dark (28°C), control, and UV exposure (29°C with 13 hours of light/11 hours of darkness), using 10-day-old Nip variety samples. For the UV treatment, we exposed the samples to 78 μ W/cm² for 20 minutes. Our findings reveal a significant increase in eccDNA numbers in leaf tissues following UV treatment, compared to both dark and control conditions (**Figure 8d**).

Notably, the eccDNA numbers in the Dark group were also higher than in the Control group. Considering that darkness represents a stress condition for rice, the resulting increase in eccDNA is reasonable. This increase induced by stress suggests that both

darkness and UV exposure can influence the growth in the number of eccDNAs. The pronounced increase in eccDNA numbers following UV exposure, however, stands out as the most significant effect observed, emphasizing UV light as a potent inducer of eccDNA formation. These observations collectively reinforce our conclusion that light exposure, including both its presence and absence, plays a role in the generation of eccDNA. We appreciate your guidance, which has been instrumental in refining our study.

Other minor points:

1. The authors should describe the data normalization method used for comparing eccDNA abundance across diverse samples. This step will enhance transparency and comprehension of the comparative analysis.

Response:

Thank you for your valuable feedback, which emphasizes the importance of a clear and standardized method for normalizing data in comparative genomic analyses.

Originally, we employed Sequencing Reads Per Million (SRPM) to quantify eccDNA abundance in our manuscript. However, we now understand that the phi29 polymerase used in Circle-Seq could lead to non-linear amplification biases. This issue was highlighted by discrepancies between SRPM values and our PCR validation results.

In response to this, we have decided to move away from using SRPM as a quantitative measure. Instead, we have adopted the Circle Score, as calculated by Circle-Map, as our primary metric. The Circle Score considers alignment quality, the length of the split segment, and the number of split reads supporting the circular DNA.

To further ensure the accuracy and comparability of our data across different samples, we propose to implement the Log_{10} for data normalization in the comparison of Circle Scores (**Figure 2d**). Utilizing \lg normalization will address potential distortions due to amplification bias and allow for a more accurate and consistent comparison of eccDNA abundance across samples. We believe this methodological refinement will significantly enhance the clarity and robustness of our comparative analysis.

2. In the bar graph, such as Fig. 1c, Fig. 2a..., the y-axis is labeled as “Numbers of eccDNA”. What does it exactly mean? The numbers represent different species of eccDNA, numbers of alignments, or numbers of reads?

Response:

Thank you for seeking clarification on the labeling of our bar graphs. We realize that the term "Numbers of eccDNA" could have been more explicitly defined, and we apologize for any ambiguity this may have caused.

In figures such as **Figure 1b** and **Figure 2a**, the y-axis labeled "Numbers of eccDNA" specifically denotes the count of distinct eccDNA species or types that have been identified within the sample. Each unit represents a unique eccDNA entity—distinct in its genomic sequence or structure from others. For example, eccDNA1, eccDNA2, and eccDNA3 would be counted as three separate entities.

To prevent any further misunderstanding, we have amended the figure label to articulate this definition precisely. The new y-axis label "Number of Detected eccDNA per Tissue" precisely communicates that the values represent the count of unique eccDNA entities identified within each specific tissue type analyzed. We hope this clarification enhances the reader's comprehension of our results and the methodology behind their presentation.

3. The gel images lack of the loading control (Fig. d-h ...).

Response:

We appreciate your attention to detail regarding the inclusion of loading controls in our gel images. We understand the necessity of loading controls in providing a reference for equal sample loading and ensuring the reliability of the observed changes in eccDNA abundance.

In response to your comment, we would like to clarify the experimental setup for the figures in question. Specifically, for Figure 4h, we included a loading control using convergent primers that amplify a region known to form eccDNA, thereby serving as an internal standard for the presence of eccDNA.

For the experiments depicted in Figures 4d-g, which involved DNA exonuclease treatments, the nature of the assay did not lend itself to traditional loading controls.

Instead, we monitored the abundance of the chromosomal linear marker *Actin1* and the mitochondrial circular marker *orf288* before and after exonuclease digestion. Notably, *orf288* was used as a comparative control, although it showed a reduction in abundance post-digestion, likely due to mechanical shearing of the larger mitochondrial DNA during the treatment process.

We would also like to emphasize that the unchanged eccDNA abundance observed in our experiments suggests that any variations are not the result of differential loading volumes. This conclusion is supported by the fact that both control markers, *Actin1* and *orf288*, displayed a decrease in abundance post-digestion, which would not be expected if the observed eccDNA stability were due to loading inconsistencies.

To further address this point, we have implemented strict protocols to ensure equal loading of samples. Each eccDNA crude sample was divided into control and digestion-treated aliquots, with equal volumes subsequently assessed by PCR. Moreover, we have meticulously minimized pipetting errors to maintain consistency across all samples.

We hope our response has clarified your doubts. Thank you for your guidance.

Reviewer #3 (Remarks to the Author):

The manuscript by Bin Han and colleagues reports on the characterisation of rice extrachromosomal circular DNA (eccDNA) in rice.

eccDNA are circular forms of DNA that have attracted quite a lot of attention recently for their implication in tumorigenesis in cancer cells (for large, Mb sized eccDNA carrying oncogenes) and for their role in retrotransposon life cycle (see recent

<https://doi.org/10.1038/s41586-023-06327-7>). In plants there are few reports of small to medium sized eccDNAs (bp to 100 kb) in glyphosate resistant plants and in a dozen of plant species with active transposable elements.

Here the authors have sequenced the eccDNA content of six rice tissues using Illumina short reads, with two biological replicates for each tissue. They show that the repertoire of eccDNA depends on the tissue with a notable hyperaccumulation of circles in leaf tissue. Some eccDNAs are validated using inverse PCR. Interestingly, the authors have detected direct repeats in some eccDNAs that could give a hint on the underlying mechanisms of formation. The study is a description of the eccDNA repertoire across development, in this respect it seems novel and well executed. Some reference to already published rice eccDNAs is lacking. Some points could be clarified so that readers have a clear overview of these eccDNA repertoires.

Major points:

1- The authors would be interested to have a look at two previous reports on rice eccDNA sequencing by the Mirouze (<https://doi.org/10.1371/journal.pgen.1006630>) and Bucher's (<https://doi.org/10.1186/s13059-017-1265-4>) labs. A putative function for a rice eccDNA originating from a LTR retrotransposon (PopRice) has even been proposed by the Cho lab (<https://doi.org/10.1093/plphys/kiad071>). Although PopRice was detected as eccDNA in the endosperm (not analyzed in this manuscript), it could be interesting to discuss these references.

Response:

Thank you for directing us to these significant studies. We have thoroughly reviewed the papers from the Mirouze, Bucher, and Cho labs, which have provided valuable insights into the role of eccDNA in rice, particularly focusing on LTR retrotransposons like

PopRice. These findings have been seamlessly integrated into our discussion, enhancing our understanding of eccDNA's function and its potential implications in plant development and stress response. This integration not only complements our findings but also broadens the context of our study. We appreciate your suggestions, as they have significantly enriched our manuscript.

2- In general, for non eccDNA specialists, it is difficult from the data presented here to appreciate the definition of the authors for a detected eccDNA. There is no visualisation of reads covering the eccDNA detected loci.

Response:

Thank you for your invaluable feedback. We understand that clarity on the definition and detection of eccDNA is crucial for all readers, not just those who specialize in this area. In response to your guidance, we have made significant enhancements to our supplementary materials and data presentation.

Firstly, we have modified the schematic diagram of circle-seq in **Figure 1a**. We now believe it more clearly assists readers in understanding our process for identifying eccDNA, particularly in how we handle split reads.

Secondly, to visually represent our findings, we have created diagrams illustrating the reads covering the detected eccDNA loci (**S_Figure 1f**). Due to the abundance of eccDNA in our samples, we selected two representative eccDNA examples and used the Integrative Genomics Viewer (IGV) for illustration.

Thirdly, we have expanded **S_Table 2-1** to include several new columns:

'All_mapped_Reads', 'Discordants', 'Split_Reads', 'Circle_Score', 'Mean_Cov', 'Std_Dev', 'Start_Cov_Inc', and 'End_Cov_Inc'. These additions provide a comprehensive overview of the sequencing data pertinent to each detected eccDNA locus. They offer a detailed breakdown of the read statistics, thereby bolstering our confidence in the identification of eccDNA.

These visualizations, alongside the enriched data provided in **S_Table 2-1**, are now included in the revised version of the manuscript. We believe these additions will greatly enhance the reader's ability to appreciate the nuances of eccDNA detection and the robustness of our methods.

3- The authors should clarify the lack of overlap between biological samples. Line 85: "Despite the large number of eccDNAs in each sample, a very small fraction was found to be identical, even between two samples derived from the same tissue." Could the authors elaborate on this point? A Venn diagram for the detected eccDNA for each of the six tissues would help evaluating the lack of overlap between the replicates.

Response:

Thank you for your comment regarding the clarification on the lack of overlap between biological samples as mentioned on line 85. We acknowledge the importance of providing a more detailed explanation on this observation, which is indeed intriguing considering the identical tissue origins of the samples.

To address this point, we have conducted a thorough analysis and have included a Venn diagram that visualizes the detected eccDNA across the six different tissues examined. This diagram is presented in **Figure 1c**, illustrates the unique and shared eccDNA among the replicates.

Upon closer examination, it becomes evident that the diversity of eccDNA profiles is substantial even among samples from the same tissue type. This could be attributed to a variety of biological factors, including but not limited to, the stochastic nature of eccDNA formation, tissue heterogeneity at the molecular level, and the sensitivity of our detection methods. These factors may contribute to the unique eccDNA landscape observed in each sample.

Furthermore, **S_Figure 1e** offers a comprehensive representation of the eccDNA repeat identification across the samples, underscoring the complexity of the eccDNA profiles and providing a clearer understanding of the extent of overlap and uniqueness.

We believe that these additions significantly enhance the manuscript by clarifying the observed phenomena and enriching the discussion with concrete visual evidence. We hope that this addresses your concerns and aids in the evaluation of the replicability and specificity of eccDNA detection across different biological samples.

4- The authors conclude about the randomness of eccDNA generation however some eccDNAs seem to be detected in more than one tissue by inverse PCR. What is the percentage of ubiquitous eccDNAs? What is the percentage of eccDNA found in only one replicate in one tissue?

Response:

We appreciate your inquiry regarding the distribution of eccDNA across various tissues and their replicates. In response, we have conducted a comprehensive analysis to quantify the ubiquity of eccDNA within our samples. To convey these findings clearly, we have included a new sheet (S_Table 2-2, S_Table 2-3 and S_Table 2-4) in our manuscript that details the percentages of eccDNAs that are ubiquitous—detected in more than one sample—as well as those that appear exclusively in a single replicate of one sample. The table presents a breakdown as follows, with 'Ubiquitous eccDNA' indicating eccDNAs found in multiple samples, 'Unique eccDNA' representing those found only in one sample.

All	Detected eccDNA	Unique eccDNAs %	Ubiquitous eccDNAs %
1wRoot-1	246	93.90	6.10
1wRoot-2	439	96.58	3.42
GE-1	130	93.08	6.92
GE-2	252	96.83	3.17
leaf-1	13630	99.80	0.20
leaf-2	8553	99.70	0.30
panicle-1	318	91.82	8.18
panicle-2	153	96.73	3.27
sheath-1	691	97.25	2.75
sheath-2	572	96.85	3.15
stem-1	411	97.57	2.43
stem-2	323	95.05	4.95

This additional data provides a comprehensive view of eccDNA distribution patterns, enhancing the transparency and depth of our comparative analysis. We trust that these details effectively address your questions and contribute to a more nuanced understanding of eccDNA dynamics in rice tissues.

5- The observation that 2% of all eccDNA carry direct repeats is of high interest. It is reminiscent of microDNAs in mammals (<https://doi.org/10.1126/science.1213307>). This reference could be discussed.

Response:

We appreciate the constructive suggestion to compare our findings with relevant research on microDNAs in mammals. After a thorough examination of the article you referenced, we have enriched our discussion to draw parallels and highlight differences. I hope that this enriched discussion will address your comment and appreciate the depth that this comparative analysis adds to our study.

6- The methods clearly indicate how "high and low confidence" were detected, however it could be helpful to have a brief sentence summarizing the definition of these two classes in the text.

Response:

We appreciate your suggestion to clarify the criteria used to classify eccDNA confidence levels in our study. Upon reflection, we have decided to streamline our approach for the sake of clarity and have updated the manuscript accordingly.

Rather than categorizing eccDNAs into 'high' and 'low' confidence, we have adopted a more rigorous and transparent criterion. We now utilize bedtools to identify eccDNAs that are consistently detected by both Circle-Map and ecc_finder software with an overlap rate exceeding 90%. This threshold was carefully chosen to balance the considerations of eccDNA length variability and the number of occurrences (**S_Figure 1a ~S_Figure 1c**).

This revised methodological description resolves any ambiguity and provides the reader with a clear understanding of how eccDNA was quantified and validated in our research.

7- Line 94: Why were the two biological replicates pooled?

Response:

Thank you for your query regarding the pooling of two biological replicates.

Upon careful examination, we observed that eccDNAs in samples from the same tissue showed a high degree of variability, as evidenced by our Venn diagrams (**Figure 1c**). Our study primarily focuses on comparing eccDNA across different tissues rather than highlighting intra-tissue variability. With this perspective in mind, we chose to pool the two biological replicates from the same tissue. We believed this strategy would provide a

more robust representation of eccDNA characteristics at the tissue level, thereby aligning more closely with our research objectives.

This approach was adopted with the intention of enhancing our understanding of eccDNA distribution and properties across various tissues. We hope this methodology will contribute valuable insights to the field and invite any further suggestions or recommendations you might have on this aspect.

8- Figure 2d: some eccDNAs have a high coverage in terms of split reads, could the authors detail them, please? Do they originate from transposable elements, from genes or from intergenic sequences?

Response:

Thank you for your advice. However, we are currently not evaluating eccDNA based solely on single split reads. Following the valuable guidance of Reviewer 1, we recognized the limitations of Circle-Seq related to the non-linear amplification characteristic of the phi29 polymerase. This non-linearity can indeed affect the quantitative assessment of eccDNA abundance. To mitigate the potential distortion caused by amplification bias, we now have adopted the Circle Score as an evaluative metric. The Circle score, calculated by Circle-Map, is an additive scoring scheme that takes into account the alignment quality, length of the split segment, and the number of split reads supporting the circular DNA.

Subsequently, we have sorted eccDNAs into two distinct categories based on their Circle Scores: 'High Circle Score eccDNAs' with a score above 100,000, and 'Low Circle Score

eccDNAs'. This stratification allowed us to analyze and compare the genomic origins of eccDNAs from both categories.

Our updated analysis, now included in the new Figure, indicates that Low Circle Score eccDNAs exhibit a pronounced enrichment in genic regions. However, when examining the High Circle Score eccDNAs, we did not find a significant correlation with any particular genomic features, be it transposable elements, genes, or intergenic sequences.

The insights garnered from this revised analysis indicate these particular High Circle Score eccDNAs do not exhibit a preference for any specific genomic region. Their distribution across various genomic areas shows no significant difference from that of eccDNAs with low Circle scores, as illustrated in S_Figure 5. This lack of a clear pattern suggests the complexity of eccDNA formation and hints at a broad spectrum of potential sources within the genome.

We have incorporated it as a supplementary figure in our manuscript. We believe that this additional information satisfactorily addresses your queries and enhances the depth of the discussion regarding our results.

9- From the methods it is not clear how many plants were pooled for each sample and what amount of genomic DNA was treated for each sample.

Response:

We acknowledge the need for precise details regarding the sample preparation in our methods section and thank you for your attention to this matter. To address your question

and improve the clarity of our experimental procedures, we have updated the methods section of our manuscript with the following information:

" For the extraction of eccDNA samples, our starting material was 100 mg of tissue powder from the Nipponbare cultivar. We prepared this tissue powder by grinding the samples under conditions of liquid nitrogen."

This addition specifies both the amount of tissue used and the rice variety, providing transparency and allowing for reproducibility of our study. We believe that this detailed description will facilitate a better understanding of our experimental setup and the scale of our eccDNA analysis.

10- While the number of split reads is given (SRPM), the mean coverage for eccDNA (total read coverage) seems to be lacking.

Response:

We acknowledge the importance of providing a complete dataset that includes not only the number of split reads but also the mean coverage for eccDNA to present a comprehensive view of our sequencing data. In light of your comment, we have enriched the **S_Table 2-1** with additional columns that detail the total read coverage for each eccDNA.

The newly added columns include:

- "All_mapped_Reads" which indicates the total number of reads that align to the eccDNA loci,
- "Mean_Cov" which represents the average depth of coverage across the eccDNA,
- "Std_Dev" (standard deviation) which provides insights into the variability of the coverage,
- "Start_Cov_Inc" and "End_Cov_Inc" which offer data on the increase in coverage at the start and end of the eccDNA, respectively.

These additions will allow for a more nuanced understanding of eccDNA abundance and provide an indicator of sequencing depth and coverage consistency across the detected eccDNAs.

I hope that this augmented data in **S_Table 2-1** will address your point and enhance the clarity of our methods and results.

11- It is not very clear whether genes overlap with eccDNA, and which genes.

Response:

We greatly appreciate your request for clarification regarding the overlap between genes and detected eccDNA within our study. To address this, we have curated a new supplementary table that meticulously documents all instances where genes coincide with identified eccDNA regions.

The newly created **S_Table 4** enumerates the genes with at least a 50% overlap with eccDNA, and annotates the gene IDs along with their corresponding genomic coordinates, as well as the sample sources of the eccDNA. This enables us to promptly and clearly discern which genes are associated with eccDNA.

This enhancement to our supplementary materials is aimed at providing readers with direct access to the specific genomic intersections between eccDNA and gene regions. We believe this level of detail will facilitate a deeper exploration of the potential functional implications of eccDNA within the genomic landscape.

We hope this additional resource effectively resolves your query and underscores our commitment to transparency and thoroughness in our research communication.

12- The title could be misleading: what is the adaptation shown in the manuscript?

Response:

Thank you for pointing out the potential for misinterpretation of our title. The term "adaptation" in our manuscript is intended to reflect the broader biological concept that the presence and dynamics of eccDNA contribute to the genomic plasticity, which can play a role in the organism's ability to respond to environmental pressures and evolutionary processes.

We have observed eccDNA formation as a universal, random event, which suggests a shared mechanism that could be utilized by rice, and potentially other organisms, as a means of genomic adaptation. The discovery of a critical role for direct repeats in the

genesis of eccDNA further supports this, indicating a potential mechanism for genetic diversity. Additionally, the tissue-specific proliferation of eccDNA in response to environmental stressors like intense light exposure suggests that eccDNA formation may be part of the plant's adaptive response to such conditions.

I hope that this revision will address your concern.

13- The cited github does not seem to exist yet. https://github.com/YxZhang-98/EccDNA_Analysis

Response:

Thank you for bringing this to our attention. We have addressed the issue with the previously cited GitHub repository link, which was indeed not accessible. The correct and updated GitHub repository, containing all the necessary scripts and data for eccDNA analysis, is now publicly available at the following URL: <https://github.com/YxZhang-XHCY/eccToolkit>.

We have taken this opportunity to ensure that the repository is well-documented and organized, facilitating ease of access and reproducibility of our computational analyses for peers in the research community. We apologize for any inconvenience the initial oversight may have caused and appreciate your patience as we rectified this matter.

We are committed to maintaining transparency and open science principles, and as such, we invite further scrutiny and use of our publicly shared tools and data. We believe that this will not only corroborate the robustness of our findings but also contribute to the advancement of the field.

Reviewers' Comments:

Reviewer #1:

Remarks to the Author:

In the revised version of the manuscript titled "The Adaptational Dynamics of Extrachromosomal Circular DNA in Rice" the authors of the study address all my major concerns. The manuscript is pleasant to read, and the experiments and the results are well described. More importantly, although I still find the study descriptive, the scientific claims of the authors are supported by evidence in the result section. I would like to congratulate the authors on their work, and personally find their results on increased eccDNA in leaves very interesting.

With that being said, I have some minor comments to some details in the manuscript that could be improved:

1- The authors conclude that rice leaves have more eccDNA and that those have lower quality, as determined by the Circle-Map circle score. I think this is incorrect. The circle quality is a weighted sum of the length and mapping qualities of discordant and split reads. It was developed to provide a single-number measure of the breakpoint read support of a circle. It is therefore a technical measure that should have nothing do with a biological process. In other words, I think the authors should not conclude that rice leaves are more prone to contain low-quality circles.

I would have found this low-quality finding if the authors would have not provided good data analysis controls. However, the authors filter by sequencing coverage and structural variant reads and show that samples achieve saturation. Yielding what I think it is a good set of circle calls. I therefore suggest the authors to report this finding as a supplement – it might indicate a technical issue in the software or their experimental procedures – and to withdraw their biological conclusions that rice leaves have low quality circles.

2- I assume the saturation curves are obtained using the same filtering scheme as the final data used in the manuscript. However, this is not indicated in the methods. Please, indicate that.

3- In the method section "paired-end sequencing for eccDNA identification" the authors write "strong and odd disagreements between the number of split reads and discordant reads were discarded". This is very unspecific. Please provide details to this.

4- This is an editorial comment rather than a scientific one. Some of the supplementary figures are not mentioned in a numerical order. For example, Supplementary Figure 1F is mentioned before Supplementary Figure 1A. This should be corrected for a smooth read.

5- Although the manuscript reads well, I have the feeling that authors are not native English speakers. I encourage the editor to help the authors address this, if possible.

Best wishes,

Iñigo Prada-Luengo

Reviewer #2:

Remarks to the Author:

I would like to thank the authors for their efforts to address my previous comments and appreciate the new tests with TEM and the inclusion of Figure 8d. These changes have addressed most of my concerns. I currently have no further questions regarding the manuscript.

Reviewer #3:

Remarks to the Author:

Dear Authors,

I appreciated your addressing my questions in great details. The general context of the eccDNA research field is discussed now in more details in the manuscript.

I am still not convinced by the term "adaptational" in the title but it is more of an opinion than a scientific comment.

Best regards

Responses to the reviewers

REVIEWERS' COMMENTS

Reviewer #1 (Remarks to the Author):

In the revised version of the manuscript titled “The Adaptational Dynamics of Extrachromosomal Circular DNA in Rice” the authors of the study address all my major concerns. The manuscript is pleasant to read, and the experiments and the results are well described. More importantly, although I still find the study descriptive, the scientific claims of the authors are supported by evidence in the result section. I would like to congratulate the authors on their work, and personally find their results on increased eccDNA in leaves very interesting.

With that being said, I have some minor comments to some details in the manuscript that could be improved:

1- The authors conclude that rice leaves have more eccDNA and that those have lower quality, as determined by the Circle-Map circle score. I think this is incorrect. The circle quality is a weighted sum of the length and mapping qualities of discordant and split reads. It was developed to provide a single-number measure of the breakpoint read support of a circle. It is therefore a technical measure that should have nothing do with a biological process. In other words, I think the authors should not conclude that rice leaves are more prone to contain low-quality circles.

I would have found this low-quality finding if the authors would have not provided good data analysis controls. However, the authors filter by sequencing coverage and structural variant reads and show that samples achieve saturation. Yielding what I think it is a good set of circle calls. I therefore suggest the authors to report this finding as a supplement – it might indicate a technical issue in the software or their experimental procedures – and to withdraw their biological conclusions that rice leaves have low quality circles.

PS: Reviewer #1 clarifies that his/her comment “I would have found this low-quality finding if the authors would have not provided good data analysis controls” should be read as “I would have found this low-quality finding **concerning** if the authors would have not provided good data analysis controls.”

Response:

It's constructive feedback. We acknowledge that the circle quality score is a technical metric aimed at assessing circle breakpoint read supported by considering the lengths and mapping qualities of discordant and split reads. It is important to note that this metric does not directly relate to biological processes. As a result, we have taken your suggestion into account and removed the statement "rice leaves are more prone to contain low-quality circles" from our manuscript. We appreciate your insightful comments and contribution to the improvement of our work.

2- I assume the saturation curves are obtained using the same filtering scheme as the final data used in the manuscript. However, this is not indicated in the methods. Please, indicate that.

Response:

Thanks for your suggestion. We have added the indication in the methods section that the saturation curves were obtained using the same filtering scheme as the final data used in the revised manuscript.

3- In the method section “paired-end sequencing for eccDNA identification” the authors write “strong and odd disagreements between the number of split reads and discordant reads were discarded”. This is very unspecific. Please provide details to this.

Response:

Yes, it's a valuable suggestion. For describing this part of the content in a specific and reliable way, we have revised the corresponding section in the methods through removing

the previously mentioned content and replacing it with specific parameters, such as Discordants ≥ 1 , Split reads ≥ 2 , and Circle score ≥ 50 .

4- This is an editorial comment rather than a scientific one. Some of the supplementary figures are not mentioned in a numerical order. For example, Supplementary Figure 1F is mentioned before Supplementary Figure 1A. This should be corrected for a smooth read.

Response:

Thank you for pointing it out. We have now re-ordered the figures to ensure they are presented in the correct numerical sequence, facilitating a smoother reading experience for the reader.

5- Although the manuscript reads well, I have the feeling that authors are not native English speakers. I encourage the editor to help the authors address this, if possible.

Response:

We acknowledge that we are not native English speakers. In our subsequent work, we will collaborate closely with the editor to improve the linguistic quality of our manuscript, ensuring clarity and precision in its expression.